# Sample Complexity Bounds for Active Ranking from Multi-wise Comparisons

**Wenbo Ren**
Dept. Computer Science & Engineering
The Ohio State University
`ren.453@osu.edu`

**Jia Liu**
Dept. Electrical & Computer Engineering
The Ohio State University
`liu.1736@osu.edu`

**Ness B. Shroff**
Dept. Electrical & Computer Engineering and Computer Science & Engineering
The Ohio State University
`shroff.11@osu.edu`

## Abstract

We study the sample complexity (i.e., the number of comparisons needed) bounds for actively ranking a set of $n$ items from multi-wise comparisons. Here, a multi-wise comparison takes $m$ items as input and returns a (noisy) result about the best item (the winner feedback) or the order of these items (the full-ranking feedback). We consider two basic ranking problems: top-$k$ items selection and full ranking. Unlike previous works that study ranking from multi-wise comparisons, in this paper, we do not require any parametric model or assumption and work on the fundamental setting where each comparison returns the correct result with probability $1$ or a certain probability larger than $\frac{1}{2}$. This paper helps understand whether and to what degree utilizing multi-wise comparisons can reduce the sample complexity for the ranking problems compared to ranking from pairwise comparisons. Specifically, under the winner feedback setting, one can reduce the sample complexity for top-$k$ selection up to an $m$ factor and that for full ranking up to a $\log m$ factor. Under the full-ranking feedback setting, one can reduce the sample complexity for top-$k$ selection up to an $m$ factor and that for full ranking up to an $m \log m$ factor. We also conduct numerical simulations to confirm our theoretical results.

## 1   Introduction

### 1.1   Background and motivation

Ranking from comparisons is a class of fundamental problems that underpin many areas in machine learning, and has found various applications in problems involving crowd-sourcing, social choices, recommendation, and searching. In such ranking problems, there is a hidden ranking among multiple items to be recovered, where items may refer to candidates, products, movies, advertisements, etc. In this paper, we study ranking from multi-wise comparisons. A multi-wise comparison refers to a query on $m$ items about the most preferred one (the winner feedback) or the full ranking (the full-ranking feedback) of these items. These comparisons may be deterministic or non-deterministic (i.e., noisy or they may return incorrect results). The noise comes from the uncertain nature of humans, the lack of information, or the underlying physics. In this paper, we focus on two goals. One is to find the top-$k$ items (ranking or ordering these items are not necessary), and the other is to find the full ranking.

Our focus is on *active ranking* (e.g., [4–7, 10, 17–19]), where "active" means that after each comparison, the learner can adaptively choose the next items to be compared according to past observations

35th Conference on Neural Information Processing Systems (NeurIPS 2021).

and comparison results. Active ranking can be viewed as active learning for ranking problems and the comparisons refer to the samples. The opposite of active ranking is passive ranking, where the learner first obtains a set of comparison results and then recovers a ranking from there. Active ranking can greatly reduce the sample complexity in many scenarios, e.g., if all comparisons return correct results with probability $\frac{2}{3}$, passively ranking $n$ items needs $\Omega(n^2)$ comparisons [3], while active ranking only needs $O(n \log n)$[1] comparisons [9, 18].

Most existing works have focused on ranking from pairwise comparisons. In contrast, we focus on ranking from *multi-wise* (or $m$-wise) comparisons. The pairwise comparisons can be viewed as multi-wise comparisons with $m = 2$. One motivation is that in many scenarios, multi-wise comparisons are more common. For instance, in video streaming websites or e-shopping apps, customers or users are normally presented more than two options, and the choices made by users can be viewed as multi-wise comparisons that suggest their preferences over these options. Studying ranking from multi-wise comparisons is useful for these types of applications. Besides, ranking from multi-wise comparisons may also reduce the cost of the learning process. In some applications, conducting the comparisons may be expensive. For instance, to find the candidates that are most preferred by the voters, people may need to do a series of surveys. Each survey contains a query about the preference order of a voter. In this application, the cost of finding the voter and asking this voter to fill the survey could far outweigh the cost of filling the survey itself. This means that the cost of conducting a multi-wise comparison is almost the same as that for a pairwise comparison, and how to reduce the number of samples by multi-wise comparisons becomes more interesting. Thus, it is not only interesting but also significant to study whether and to what degree we can reduce the sample complexity for ranking by using multi-wise comparisons.

We focus on a non-parametric model, where each comparison returns the correct result with a certain probability $q > \frac{1}{2}$ and an arbitrary incorrect result otherwise. When $q = 1$, the comparisons are deterministic and always return correct results, and when $q < 1$, we say the comparisons are non-deterministic or noisy. This differs from parametric models[2] that assume that each item holds a value representing the users' preference on this item. In parametric models, the comparisons may provide more information than the setting in this paper[3], and thus, the conclusion drawn under parametric models cannot be directly applied to this paper.

## 1.2 Problem formulation

Assume that there are $n$ items, indexed by $1, 2, 3, ...n$, that form the item set $[n]$[4]. We further assume that these items have a *unique unknown true* ranking $r_1 \succ r_2 \succ \cdots \succ r_n$, where for any items $i$ and $j$, notation $i \succ j$ means that item $i$ ranks higher (or is more preferred) than item $j$. Assume that we can compare at most $m$ items at a time. The comparisons can be either deterministic or non-deterministic. We also assume that the comparisons are independent across time, items, and sets, which is standard in the literature (e.g., [5–7, 10, 11, 13, 17–22]). Here, we note that the independence is based on the assumption that the hidden parameters and ranking are some fixed values.

When the comparisons are deterministic, the comparisons always return the correct results, i.e., the best item under the winner feedback model or the true ranking of the compared items under the full-ranking feedback model. In the deterministic case, our goal is to find the exact top-$k$ items or the true ranking.

When the comparisons are non-deterministic, we assume that they return the correct results with a certain fixed probability $q \in (\frac{1}{2}, 1)$. In this case, it is infeasible to rank the items with 100% confidence, and thus, our goal is to find the ranking with confidence $1 - \delta$ for some $\delta \in (0, \frac{1}{2})$. We focus on the case where $q = \frac{2}{3}$. We note that this does not lose much generality. When $q > \frac{2}{3}$, the

---

[1]All log in this paper are natural $\log$ unless explicitly noted.

[2]For instance, parametric models can be the Bradley-Terry-Luce (BTL) model [2], the Plackett-Luce (PL) model [15], or the multinomial logit (MNL) model [14].

[3]For instance, given three items $i, j, k$ with $i$ being the best and $k$ being the worst, in the setting of this paper with $q = 0.7$, the multi-wise comparison returns the best item $i$ with probability 0.7, and return $j$ or $k$ with some unknown probabilities, which does not provides information about the orders of $j$ and $k$. In contrast, in a parametric model, the comparison over $\{i, j, k\}$ would return $i, j$, or $k$ with probability 0.7, 0.2, and 0.1, respectively, which not only provides information about the best one but also information for ordering $j$ and $k$.

[4]For any positive integer $l$, we define $[l] := \{1, 2, 3, ..., l\}$.

algorithms and sample complexity bounds in this paper can be directly applied. When $q < \frac{2}{3}$, we can use repeated comparisons to simulate one comparison with correct probability at least $\frac{2}{3}$. To see this, consider a set over which a comparison returns the correct result with probability $\frac{1}{2} + \Delta > \frac{1}{2}$. By comparing this set for $\lceil \frac{1}{2\Delta^2} \log 3 \rceil$ times, the item or permutation that occurs most often is the correct result with probability at least $\frac{2}{3}$ (by Hoeffding Inequality). Thus, we can use the above method to substitute the comparisons in the algorithms designed for the case where $q = \frac{2}{3}$ while only introducing an additional $\frac{1}{\Delta^2}$ factor on the sample complexity.

This $(\frac{1}{2} + \Delta)$ $m$-wise comparison model can also be justified in many scenarios that use an iterative subroutine to conduct $m$-wise comparisons over a set in a smaller time-scale. Here, we give a simple example. We use $p_{i,S}$ to denote the probability that item $i$ wins the comparison performed on set $S$ (assuming $|S| \leq m$) and we use $i^*$ to denote the best item of $S$. If $p_{i^*,S} \geq p_{i,S} + \Delta$ for any item $i \neq i^*$ and any set $S$, then by repeatedly comparing $S$ for $\Theta(\frac{1}{\Delta^2} \log |S|)$ times, one can find the best item of $S$ with confidence $\frac{2}{3}$. We can use the above procedure as a subroutine in each iteration of our algorithms, and get the algorithms for this case while only introducing an additional $\log m$ factor to the sample complexities. In more general cases, we can use similar tricks but the additional factors may vary.

## 1.3 Main results

The main results of this paper are summarized in Table 1. We note that due to space limitation, all proofs in this paper are left to the supplementary material.

Table 1: Main results of this paper. All results are established in this paper unless explicitly cited. For results of non-deterministic feedback, $\delta$ is the error probability and we assume that all comparisons return correct results with probability $\frac{2}{3}$.

| Problem | | Top-$k$ Selection | Full Ranking |
|---|---|---|---|
| Winner Feedback Model | Deterministic Feedback | $\Theta(\frac{n}{m} + k)$ | $\Theta(\frac{n \log n}{\log m})$ [17] |
| | Non-Deterministic Feedback | $O((\frac{n}{m} + k) \log \frac{k \log m}{\delta})$ $\Omega(k + \frac{n}{m} \log \frac{k}{\delta})$ | $\Theta(n \log \frac{n}{\delta})$ |
| Full-Ranking Feedback Model | Deterministic Feedback | $\Theta(\frac{n}{m})$ | $\Theta(\frac{n \log n}{m \log m})$ |
| | Non-Deterministic Feedback | $O(\frac{n}{m} \log \frac{\min\{n/m,k\} \log m}{\delta})$ $\Omega(\frac{n}{m} \log_m \frac{k}{\delta})$ | $O(\frac{n}{m} \log \frac{n}{m\delta})$ $\Omega(\frac{n}{m} \log_m \frac{n}{\delta})$ |

## 2 Related works

When $m = 2$ and the comparisons are deterministic, the top-$k$ selection problem becomes the classical pairwise $k$-selection problem, which requires $\Theta(n)$ comparisons [1], and the full ranking problem becomes the classical pairwise sorting problem, which requires $\Theta(n \log n)$ comparisons. Thus, ranking from multi-wise comparisons can also be viewed as extensions of these foundational problems. Surprisingly, these extensions have not been well understood.

For top-$k$ selection from non-deterministic pairwise comparisons, the authors of [9] showed that $\Theta(n \log \frac{k}{\delta})$ comparisons are necessary and sufficient to reach confidence level $1 - \delta$. For probably approximately correct (PAC)[5] top-$k$ selection with error tolerance $\epsilon > 0$, the authors of [5, 7] proved a $\Theta(\frac{n}{\epsilon^2} \log \frac{1}{\delta})$ bound for $k = 1$, and the authors of [16, 18] proved a $\Theta(\frac{n}{\epsilon^2} \log \frac{k}{\delta})$ bound for $k \leq \frac{n}{2}$.

For full ranking from non-deterministic pairwise comparisons, an early work was [9], whose authors proved that when the comparisons of all pairs have the same noise level, then to get the full ranking with confidence $1 - \delta$, $\Theta(n \log \frac{n}{\delta})$ comparisons are necessary and sufficient. The authors of [17]

---

[5]The PAC setting means that we want to find the ranking approximately with an error tolerance $\epsilon$ and a confidence $1 - \delta$, where the $\epsilon$ tolerance means that for any two items $i$ and $j$, item $i$ is viewed to rank higher than item $j$ if item $i$ wins the comparisons over item $j$ with probability $\frac{1}{2} - \epsilon$ or higher.

extend the above results to the case where the comparisons on different pairs have different noise levels, and also obtained the $\Theta(n \log \frac{n}{\delta})$ bounds. The authors of [5–7] showed the $\Theta(\frac{n}{\epsilon^2} \log \frac{n}{\delta})$ order-wise tight upper and lower bounds for PAC full ranking with error tolerance $\epsilon > 0$.

Thus, we can see that for $m = 2$ (i.e., ranking from pairwise comparisons), people have already established optimal sample complexity bounds for top-$k$ selection and full ranking. However, the corresponding multi-wise ranking problems have been relatively under-explored. Most of these works have focused on parametric models such as the PL model and the MNL model. Under the MNL model, the work in [4] showed that under certain cases, one can get an $m$-reduction from $m$-wise comparisons over ranking from pairwise comparisons. However, for most cases, using $m$-wise comparisons does not reduce the sample complexity (ignoring constant factors). For multi-wise full ranking under the MNL model, the works in [16, 17] showed that the lower bound is the same as that using pairwise comparisons, i.e., there is no reduction by using $m$-wise comparisons. Under the list-wise PL model with certain types of feedback, we may achieve up to $m$-reduction in the sample complexity according to [19].

In contrast to the above works, we do not assume any parametric model, and instead use a non-parametric model where each comparison returns a correct result with a certain fixed probability $q$. As noted in Section 1.1, the results drawn from the parametric models cannot be directly applied to this non-parametric model. Multi-wise ranking under this non-parametric setting is even less under-explored in the literature. The most related work to this paper is [17], where the authors showed a $\Theta(\frac{n \log n}{\log m})$ bound for full ranking from the winner feedback, which solves one of the eight cases (top-$k$ selection or full ranking, winner feedback or full-ranking feedback, deterministic feedback or non-deterministic feedback) that is studied in this paper and shown in Table 1. We will study the rest seven cases in this paper.

## 3  Top-$k$ selection from winner feedback

### 3.1  Deterministic feedback

**Lower bound.** We first state an $\Omega(\frac{n}{m} + k)$ lower bound in Theorem 1. When $m = 2$, this lower bound reduces to $\Omega(n)$, the same as that for the pairwise $k$-selection problem.

**Theorem 1.** *To find the top-$k$ items from $n$ items by $m$-wise deterministic winner feedback, any algorithm needs to conduct at least $\Omega(\frac{n}{m} + k)$ comparisons.*

**Upper bound** When $m = 2$, the problem reduces to the basic pairwise $k$-selection problem, which requires $\Theta(n)$ comparisons. One algorithm to solve the pairwise $k$-selection problem is Quick-Select (QS) [12]. With $n$ items, QS randomly chooses a pivot, splits other items into two piles based on whether them are larger or smaller than the pivot, and then recursively calls QS on one of these two piles according to the piles' sizes. The expected number of comparisons required by QS is $O(n)$ (in the worst case it will be $O(n^2)$).

However, if we want to get the $O(\frac{n}{m} + k)$ sample complexity upper bound for multi-wise top-$k$ selection that matches the lower bound stated in Theorem 1, we cannot split the items into two piles because splitting $(n - 1)$ items into two piles requires $\Omega(n)$ comparisons. Instead, we split them into $m$ piles that are formed by $(m - 1)$ randomly chosen pivots. Also, if the algorithm splits all the non-pivot items, then we still have $\Omega(n)$ sample complexity, which is sub-optimal. Our key idea is to stop splitting if we have identified the elements of the first several piles and the number of items in these piles along with the corresponding pivots is no less than $k$. By analyzing how the items will be split, we will show that we only need to conduct $O(\frac{n}{m} + k)$ comparisons in expectation before terminating splitting. After the splitting, we only need to focus on the pile that contains the $k$-th item as the piles that rank higher contain only items better than the $k$-th item and the others contain only items worse then the $k$-th item. We can show that this pile is of size $O(\frac{n}{m})$ in expectation, and thus, finding the top-$k$ items from it only takes $O(\frac{n}{m})$ comparisons by QS. Therefore, we find the top-$k$ items by using $O(\frac{n}{m} + k)$ comparisons in expectation.

We name the above method as Multi-wise Quick-Select (MQSelect) and describe it in Algorithm 1. The theoretical performance of MQSelect is formally stated in Theorem 2.

**Theorem 2.** *MQSelect returns the top-k items of $S$ after $O(\frac{n}{m} + k)$ comparisons in expectation.*

---

**Algorithm 1** Multi-wise Quick-Select$(S, m, k)$ (MQSelect).

---

1: **if** $|S| \leq m$ **then**
2:     Compare $S$ for $k$ times; After each comparison remove the winner from $S$ and add it to $Ans$;
3:     **return** $Ans$;
4: **end if**
5: Randomly choose $m - 1$ items and form a pivot set $V$;
6: $R_1 \leftarrow S - V$; $R_2, R_3, ..., R_m \leftarrow \emptyset$; $D_0 \leftarrow \emptyset$; $A_i \leftarrow \emptyset$ for $i \in [m - 1]$;
7: **for** $t = 1, 2, 3, ..., m - 1$ **do**
8:     Compare $V$ once and denote the winner as $v_t$;
9:     $V \leftarrow V - \{v_t\}$; $E \leftarrow \emptyset$;
10:     **while** $R_t \neq \emptyset$ **do**
11:         Choose items from $R_t$ and add them to $E$ until $|E|$ reaches $\min\{m - 1, |R_t|\}$;
12:         Compare $E \cup \{v_t\}$ and denote the winner as $w$;
13:         **if** $w = v_t$ **then**
14:             $R_{t+1} \leftarrow R_{t+1} + E$; $R_t \leftarrow R_t - E$; $E \leftarrow \emptyset$;
15:         **else**
16:             $A_t \leftarrow A_t + \{w\}$; $R_t \leftarrow R_t - \{w\}$; $E \leftarrow E - \{w\}$;
17:         **end if**
18:     **end while**
19:     $D_t \leftarrow D_{t-1} \cup A_t \cup \{v_t\}$;
20:     **if** $|D_t| = k$ **then**
21:         **return** $D_t$;
22:     **else if** $|D_t| = k + 1$ **then**
23:         **return** $D_t - \{v_t\}$;
24:     **else if** $|D_t| > k + 1$ **then**
25:         **return** $D_{t-1} \cup$ Quick-Select$(A_t, k - |D_{t-1}|)$;
26:     **end if**
27: **end for**
28: **return** $D_{m-1} \cup$ Quick-Select$(R_m, k - |D_{m-1}|)$;

---

According to the lower bound stated in Theorem 1, MQSelect is optimal up to a constant factor. When $m = 2$, the $\Theta(\frac{n}{m} + k)$ bound reduces to $\Theta(n)$, the same as that for pairwise $k$-selection.

## 3.2   Non-deterministic feedback

**Lower bound.** The lower bound for $m$-wise top-$k$ selection from non-deterministic winner feedback is stated in Proposition 3. Later Theorem 5 will show that this bound is optimal up to a $\log \frac{k \log m}{\delta}$ factor, and when $k \leq \frac{n}{m}$, this bound is optimal up to a $\log \log m$ factor.

**Proposition 3.** *There is an $n$-sized instance such that to find the top-$k$ items with confidence $1 - \delta$ by using $m$-wise non-deterministic winner feedback, any algorithm needs at least $\Omega(k + \frac{n}{m} \log \frac{k}{\delta})$ comparisons in expectation.*

**Upper bound.** We first introduce a simple subroutine Basic Compare (BC) in Algorithm 2. Given $\delta$

---

**Algorithm 2** Basic Compare$(S, \delta)$ (BC).

---

1: Compare $S$ for $N_0 = \lceil 18 \log \frac{1}{\delta} \rceil$ times;
2: **return** the result that is returned for the most number of times;

---

and input set $S$, BC compares $S$ for $N_0 = \lceil 18 \log \frac{1}{\delta} \rceil$ times and returns the most often result, which is correct with probability at least $1 - \delta$. This can be shown by using Hoeffding inequality

$$\mathbb{P}\{X \leq N_0/2\} \leq \exp\{-2N_0(2/3 - 1/2)^2\} \leq \delta, \tag{1}$$

where $X$ is the number of times that the correct result is returned.

We now present a relatively simple algorithm called Basic $k$-Selection (BKS), which will be used later for developing another algorithm. The idea of BKS is that we replace the $m$-wise comparisons in

MQSelect by the calls of BC with a certain confidence $1 - \delta_1$. Intuitively, we can set $\delta_1 = \frac{\delta}{n^2}$ to make BKS return the correct result with confidence $1 - \delta$ as MQSelect conducts at most $n^2$ comparisons. In this case, we will need $O((k + \frac{n}{m}) \log \frac{n}{\delta})$ comparison. However, by showing that with probability at least $1 - \delta_0$, MQSelect conducts at most $N = O((k + \frac{n}{m}) \log \frac{nk \log m}{m\delta_0})$ comparisons, we can set $\delta_1 = \frac{\delta_0}{N}$ and $\delta_0 = \frac{\delta}{3}$ and get a better upper bound $O((\frac{n}{m} + k) \log \frac{nk \log m}{m\delta})$. In fact, if we use BKS with $\delta_1 = \frac{\delta}{n^2}$, then the algorithm we construct in the later subsection will have a worse upper bound. We describe BKS in Algorithm 3. Its theoretical performance is formally stated in Lemma 4.

---

**Algorithm 3** Basic $k$-Select$(S, m, k, \delta)$ (BKS).

---

1: $\delta_0 \leftarrow \frac{\delta}{3}$; $T_1 \leftarrow 1 + \frac{n}{m-2} \log \frac{2(m-1)(m-2)}{\delta_0}$; $\delta_1 \leftarrow \frac{\delta_0}{6k+5T_1}$ if $m > 2$; $\delta_1 \leftarrow \frac{\delta_0}{|S|^2}$ if $m = 2$;
2: Run MQSelect on $M$, but using calls of BC with confidence $1 - \delta_1$ to replace multi-wise comparisons (except those in the call of QS);
3: For the call of QS, we replace the pairwise comparisons with calls of BC with confidence $1 - \frac{\delta_0}{|A_t|^2}$;

---

**Lemma 4.** *BKS terminates after $O((\frac{n}{m} + k) \log \frac{nk \log m}{m\delta})$ comparisons in expectation, and with probability at least $1 - \delta$, returns the top-$k$ items of $S$.*

With BKS, we develop an enhanced algorithm for multi-wise top-$k$ selection that can remove the $\log n$ factor in the sample complexity of BKS. At each round $t$, we split the remaining items into subsets with sizes at most $mk$. For each set, we use BKS with confidence $1 - \frac{\delta_t}{k}$ to find the top-$k$ items and call them *winners*. We can show that with probability at least $1 - \delta_t$, the winners of all subsets together contain the top-$k$ items of $[n]$. We keep the winners and remove other items. Repeat the above step on remaining items for multiple rounds until only $k$ winners remain, and these $k$ winners are the top-$k$ items of $[n]$ with probability at least $1 - \sum_{t=1}^{\infty} \delta_t$. By setting proper values of $\delta_t$, we can find the top-$k$ items with confidence $1 - \delta$ by using $O((\frac{n}{m} + k) \log \frac{k \log m}{\delta})$ comparisons. We name this algorithm Multi-wise Tournament $k$-Select (MTKS) and describe it in Algorithm 4. Its theoretical performance is formally stated in Theorem 5.

---

**Algorithm 4** Multi-wise Tournament $k$-Select $(S, m, k, \delta)$ (MTKS)

---

1: **if** $|S| \leq mk$ **then**
2:     **return** BKS$(S, m, k, \delta)$;
3: **end if**
4: Set $t \leftarrow 0$ and $R_1 \leftarrow S$;
5: **repeat**
6:     $t \leftarrow t + 1$; $\delta_t \leftarrow \frac{6\delta}{\pi^2 t^2}$;
7:     Distribute $R_t$ to $\lceil \frac{|R_t|}{mk} \rceil$ disjoint sets $A_1, A_2, ..., A_d$, each with size at most $mk$;
8:     For $i \in [d]$, let $B_i \leftarrow$ BKS$(A_i, m, k, \frac{\delta_t}{k})$;
9:     $R_{t+1} \leftarrow \bigcup_{i \in [d]} B_i$;
10: **until** $|R_{t+1}| \leq k$
11: **return** $R_{t+1}$;

---

**Theorem 5.** *MTKS terminates after $O((\frac{n}{m} + k) \log \frac{k \log m}{\delta})$ comparisons in expectation, and with probability at least $1 - \delta$, returns the top-$k$ items of $S$.*

We can see that, by using multi-wise comparisons, MTKS can achieve up to an $m$-reduction in the sample complexity. According to Proposition 3, when $k \leq \frac{n}{m}$, our upper bound is optimal up to a $\log \log m$ factor, which is almost constant for most of the practical cases.

# 4 Full ranking from winner feedback

## 4.1 Deterministic feedback

This setting has been studied in [17], where the authors have shown $\Theta(n \log_m n)$ upper and lower bounds for the sample complexity. We restate their results in Proposition 6. By using $m$-wise comparisons, we can reduce the sample complexity for finding the full ranking by a $\log m$ factor.

**Proposition 6** ([17]). *To exactly rank $n$ items by using $m$-wise deterministic comparisons under the winner feedback model, $\Theta(n \log_m n)$ comparisons are necessary and sufficient.*

## 4.2 Non-deterministic feedback

Unlike full ranking from deterministic comparisons, when the comparisons are non-deterministic, this logarithmic reduction may not exist any more. This result is stated in Theorem 7.

**Theorem 7.** *There is an $n$-sized instance such that to get the full ranking with confidence $1 - \delta$ from $m$-wise winner feedback, any algorithm needs at least $\Omega(n \log \frac{n}{\delta})$ comparisons.*

For $m = 2$, i.e., when using pairwise comparisons, the algorithms in [18] can already find the full ranking of $n$ items with confidence $1 - \delta$ by $O(n \log \frac{n}{\delta})$ comparisons, matching the lower bound in Theorem 7. Thus, we do not propose an algorithm for $m > 2$ and one can directly use the pairwise algorithms for $m > 2$ and obtain optimal sample complexity (ignoring constant factors).

# 5 Top-$k$ selection from full-ranking feedback

## 5.1 Deterministic feedback

For the lower bound, since each comparison involves at most $m$ items and all items need to be involved in at least one comparison to get the top-$k$ items, we immediately have the $\Omega(\frac{n}{m})$ lower bound. For the upper bound, we develop a Quick-Select-like algorithm MQSelect-FRF (Multi-wise Quick-Select from Full-Ranking Feedback) under the full-ranking feedback model similar to MQSelect. In the splitting, MQSelect-FRF takes less comparisons since the full-ranking feedback provides more information than the winner feedback, which removes the $O(k)$ term in the upper bound that exists in the sample complexity of MQSelect. Due to space limitation, we only state the bounds in Theorem 8 and the proofs and algorithms are relegated to the supplementary material.

**Theorem 8.** *To get the top-$k$ items of $n$ items from deterministic $m$-wise full-ranking feedback, $\Theta(\frac{n}{m})$ comparisons are necessary and sufficient.*

## 5.2 Non-deterministic feedback

**Lower bound.** In this part, we provide a lower bound in Proposition 9, which is optimal up to a $\log m \cdot \log \log m$ factor. This lower bound is proved by reducing the pairwise ranking problem to the list-wise ranking problem. In fact, we are not aware whether there is a stronger lower bound, and this problem requires future investigation.

**Proposition 9.** *There is an $n$-sized instance such that to find the top-$k$ items by $m$-wise non-deterministic full-ranking feedback, any algorithm needs $\Omega(\frac{n}{m} \log_m \frac{k}{\delta})$ comparisons.*

**Upper bound.** We develop the full-ranking algorithm following the similar steps as in MTKS. First, we develop an algorithm named BKS-FRF (BKS from Full-Ranking Feedback) similar to BKS, which replaces the comparisons of the deterministic-comparison algorithm by a call of BC with a certain confidence and has sample complexity $O(\frac{n}{m} \log \frac{n \log m}{m \delta})$. When $k < \frac{n}{m}$, we further use similar steps as in MTKS and develop an algorithm named MTKS-FRF (MTKS from Full-Ranking Feedback) that further reduces the sample complexity to $O(\frac{n}{m} \log \frac{\min\{n/m, k\} \log m}{\delta})$, which is optimal up to a $\log m \cdot \log \log m$ factor. Due to space limitation, we only state the upper bound in Theorem 10 and the algorithms are presented in the supplementary material.

**Theorem 10.** *MTKS-FRF terminates after $O(\frac{n}{m} \log \frac{\min\{n/m, k\} \log m}{\delta})$ comparisons in expectation, and with probability at least $1 - \delta$, returns the top-$k$ items of $S$.*

# 6 Full ranking from full-Ranking feedback

## 6.1 Deterministic feedback

**Lower bound.** We first show the lower bound for this setting in Theorem 11. The proof is based on an information-theoretic approach. Specifically, there are $n!$ permutations, which implies that the information entropy of the true ranking is $\log(n!)$. Each $m$-wise comparison has at most $m!$ possible results. Thus, each $m$-wise comparisons provides at most $\log(m!)$ information towards the true ranking. By Fano's Inequality [8], to get the true ranking with probability at least $\frac{3}{4}$, the information about the true ranking one needs to obtain is at least $\Omega(\log(n!))$. Since $\log(n!) = \Theta(n \log n)$ and $\log(m!) = \Theta(m \log m)$, the lower bound is at least $\Omega(\frac{n \log n}{m \log m})$.

**Theorem 11.** *To get the full ranking of $n$ items by using $m$-wise comparisons under the full-ranking feedback model, any algorithm needs at least $\Omega(\frac{n \log n}{m \log m})$ comparisons in expectation.*

---

**Algorithm 5** Multi-wise Quick-Sort$(S, m)$ (MQSort)

---

1: **if** $|S| \leq m$ **then**
2:     Compare $S$, obtain its full ranking and return;
3: **end if**
4: Randomly choose $h := \lfloor \frac{m}{2} \rfloor$ items and form pivot set $V$;
5: Compare $V$ to get its full ranking $v_1 \succ v_2 \succ \cdots \succ v_h$;
6: Let $v_0$ represent a dummy item that ranks higher than all other items;
7: $R \leftarrow S - V$; $A_i \leftarrow \emptyset$ for $i = 0, 1, 2, ..., h$;
8: **while** $R \neq \emptyset$ **do**
9:     Choose $(m - h)$ items from $R$ and form set $E$;
10:     Compare set $E \cup V$ and get the ranking;
11:     **for** item $i$ in $E$ **do**
12:         Add item $i$ to set $A_j$ if $j$ items in $V$ ranks higher than $i$ (i.e., if $v_{j-1} \succ i \succ v_j$);
13:     **end for**
14: **end while**
15: **for** $j = 0, 2, 3, ..., h$ **do**
16:     Call MQSort$(A_j, m)$ to get the full ranking of $A_j$;
17:     Insert the sorted items of $A_j$ into between $v_{j-1}$ and $v_j$;
18: **end for**
19: **return** the current ranking of $S$;

---

**Upper bound.** Since it is well-known that full ranking from pairwise comparisons needs $O(n \log n)$ comparisons, developing a full ranking algorithm under the non-deterministic full-ranking feedback model with sample complexity $O(\frac{n}{m} \log n)$ is trivial by viewing an $m$-wise comparison as $\lfloor \frac{m}{2} \rfloor$ pairwise comparisons. Instead, we propose an algorithm called Multi-wise Quick-Sort (MQSort) with expected sample complexity $O(\frac{n \log n}{m \log m})$, better than the above trivial bound and matches the lower bound stated in Theorem 11. MQSort can be viewed as an extension of the classical Quick-Sort algorithm, but requires more complicated mathematical analysis to prove its sample complexity. We describe MQSort in Algorithm 5 and formally state its theoretical performance in Theorem 12.

**Theorem 12.** *MQSort terminates after $O(\frac{n \log n}{m \log m})$ $m$-wise comparisons in expectation and returns the full ranking of $S$.*

## 6.2 Non-deterministic feedback

**Lower bound.** When the full ranking is obtained, then the top-$\frac{n}{2}$ items can also be obtained for free. Thus, we immediately have the $\Omega(\frac{n}{m} \log_m \frac{n}{\delta})$ full-ranking lower bound in Corollary 13 by invoking Proposition 9. Corollary 13 is optimal up to a $\log m$ factor. Whether a tighter bound exists remains an open problem and requires further investigation.

**Corollary 13.** *There is an $n$-sized instance such that to find the full ranking from $m$-wise full-ranking feedback with confidence $1 - \delta$, any algorithm needs $\Omega(\frac{n}{m} \log_m \frac{n}{\delta})$ comparisons in expectation.*

Next, we modify the algorithm in [9] to achieve an $O(\frac{n}{m} \log \frac{n}{m\delta})$ upper bound, which is stated in Theorem 14. Due to space limitation, we leave the algorithm to the supplementary material.

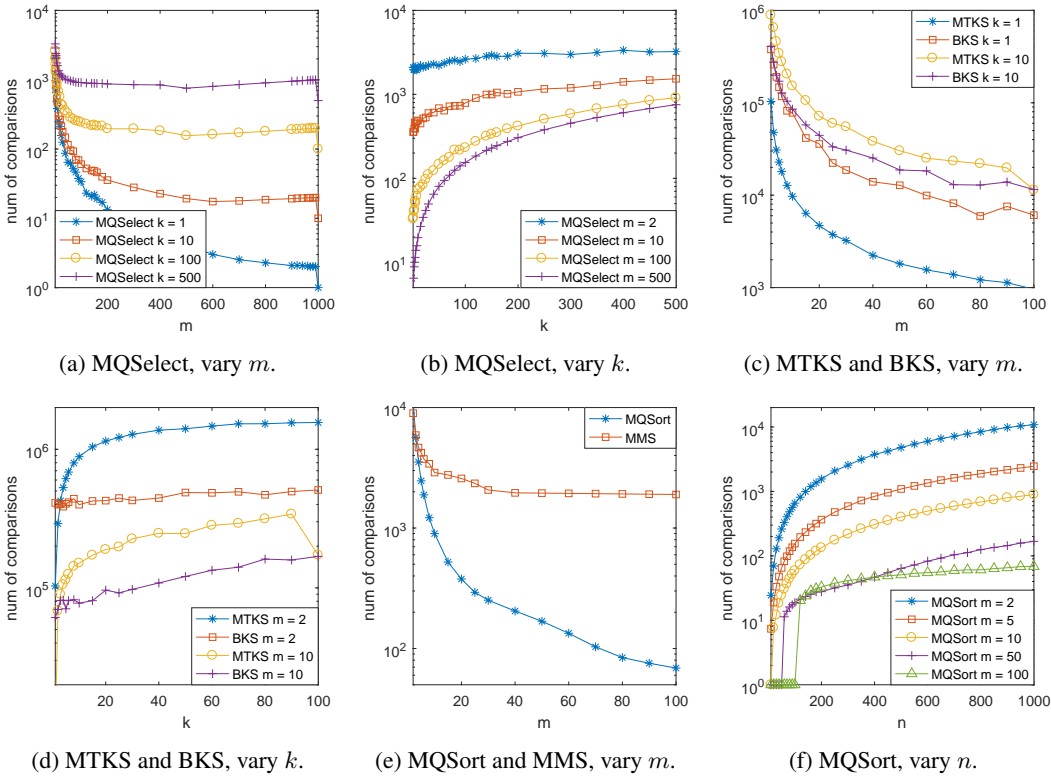

| (a) MQSelect, vary $m$. | (b) MQSelect, vary $k$. | (c) MTKS and BKS, vary $m$. |
| (d) MTKS and BKS, vary $k$. | (e) MQSort and MMS, vary $m$. | (f) MQSort, vary $n$. |

Figure 1: Performance comparisons of all algorithms: In all figures, $n = 1000$ (except (f)), $\delta = 0.01$ (if applicable), and all points are averaged over 100 independent trials with random true rankings.

**Theorem 14.** *There is an algorithm that finds the full ranking of $n$ items by $m$-wise full-ranking feedback with confidence $1 - \delta$ and conducts $O(\frac{n}{m} \log \frac{n}{m\delta})$ comparisons in expectation.*

## 7 Numerical results

In this section, we conduct numerical experiments to verify out theoretical results. The codes can be found in our GitHub repo.[6]

In Figure 1 (a,b), we present the results of MQSelect. In Figure 1 (a), we set $n = 1000$ and $k = \{1, 10, 100, 500\}$, and vary the value of $m$. In Figure 1 (b), we set $n = 1000$ and $m = \{2, 10, 100, 500\}$, and vary the value of $k$. First, from the results, we can see that that given the same value of $k$, the number of comparisons conducted by MQS decreases as $m$ increases, and the decreasing rate is approximately $\frac{1}{m}$ according to Figure 1 (a), consistent with our theory that the sample complexity of MQSelect is $O(\frac{n}{m} + k)$. This indicates that using multi-wise comparisons can significantly reduce the required number of comparisons for top-$k$ ranking, especially for a large value of $m$. Second, Figure 1 (b) shows that for a given value of $m$, when $k$ increases, the number of comparisons conducted by MQSelect increases nearly linearly. The larger the $m$-values, the closer to linear the increasing rates are, which is also consistent with the theory.

In Figure 1 (c,d), we compare the performance of MTKS and BKS. In Figure 1 (c), we set $n = 1000$, $\delta = 0.01$, and $k = \{1, 10\}$, and vary the value of $m$. In Figure 1 (d), we set $n = 1000$, $\delta = 0.01$, and $m = \{2, 10\}$, and vary the value of $k$. First, we can see from Figure 1 (c) that when $m$ increases, the number of comparisons conducted by MTKS and BKS both decrease, and the decreasing rate is larger for smaller values of $k$, which is consistent with the theory that the sample complexities of both algorithms depend on $(\frac{n}{m} + k)$. This also indicates that by using multi-wise comparisons, we can save a significant number of comparisons for top-$k$ selection. Second, from Figure 1 (c), we

---

[6]https://github.com/WenboRen/Multi-wise-Ranking.git

can see that, when $k = 1$, MTKS uses less comparisons than BKS for almost all values of $m$, which is consistent with the theory that the sample complexity of MTKS depends on $\log \frac{k \log m}{\delta}$, smaller than the $\log \frac{nk \log m}{m\delta}$ rate of BKS. However, from Figure 1 (c), we can also see that, when $k$ is larger, e.g., $k = 10$, the performance of BKS is slightly better than that of MTKS, which indicates that MTKS has a larger constant factor. Third, according to Figure 1 (d), when $k$ increases, the number of comparisons conducted by both algorithms tend to increase except for a small set of points. When $m = 1$, the increasing rate of the number of conducted comparisons with $k$ is smaller than that when $m = 10$, which is consistent with the theory that the sample complexities of both algorithms depend on $(\frac{n}{m} + k)$. Thus, when $m$ is larger, the complexity of the algorithms are more sensitive with $k$.

In Figure 1 (e,f), we present the numerical results of MQSort. To show how the full-ranking feedback can help reduce the sample complexity for finding the full ranking, we compare MQSort with Multi-wise Merge Sort (MMS) algorithm proposed in [17], which uses the winner feedback. In theory, to rank $n$ items, MQSort uses $O(\frac{n \log n}{m \log m})$ comparisons and MMS uses $O(\frac{n \log n}{\log m})$ comparisons. In Figure 1 (e), we set $n = 1000$ and vary the value of $m$. In Figure 1 (f), we set $m = \{2, 5, 10, 50, 100\}$ and vary the value of $n$. First, from Figure 1 (e,f), we can see that when $m$ increases, the number of comparisons conducted by MQSort decreases, and the decreasing rate is nearly $\frac{1}{m \log m}$, which is consistent with the theory that the sample complexity of MQSort is $O(\frac{n \log n}{m \log m})$. This indicates that by using multi-wise full-ranking feedback with large values of $m$, we can significantly reduce the sample complexity for finding the full ranking. Second, we can also see that MQSort uses less comparisons than MMS for $m \geq 5$, and the decreasing rate of the complexity is also faster than MMS, consistent with the theory that MQSort has sample complexity $O(\frac{n \log n}{m \log m})$ and MMS has sample complexity $O(\frac{n \log n}{\log m})$. This suggests that compared to winner feedback, using full-ranking feedback is more efficient in terms of sample complexity for finding the full ranking.

## 8   Conclusion

This paper studied the problems of selecting the top-$k$ items or finding the full ranking of a set of $n$ items by using $m$-wise comparisons under the winner feedback model or the full-ranking feedback model. The comparisons can be either deterministic or non-deterministic. For all eight combinations of settings (top-$k$ selection or full ranking, winner feedback or full-ranking feedback, deterministic or non-deterministic feedback), we proposed algorithms, derived upper bounds, and/or proved lower bounds. For four settings, we obtained tight upper and lower bounds (up to constant factors). For three settings, we obtained upper and lower bounds where there is only logarithmic gaps between them. The results in this paper showed that by using multi-wise comparisons, one could dramatically reduce the number of comparisons needed for the ranking problems compared to ranking from pairwise comparisons. The numerical results presented also confirmed our theoretical predictions.

### Acknowledgments and Disclosure of Funding

This work has been supported in part by NSF grants CAREER CNS-2110259, CNS-2112471, CNS-2102233, CNS-2112471, CNS-2106932, CNS-1955535, CCF-2110252, ECCS-2140277, IIS-2112471, and a Google Faculty Research Award.

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
