*Proof.* When $m = 2$, i.e., for $k$-selection from pairwise comparisons, it is well-known that one needs at least $\Omega(n)$ pairwise comparisons to find the top-$k$ items. Any $m$-wise comparison can be simulated by $(m-1)$ pairwise comparisons. Thus, by using $m$-wise comparisons, one needs at least $\Omega(\frac{n}{m})$ comparisons for finding the top-$k$ items. Otherwise, we would get a contradiction against the lower bound for top-$k$ items selection from pairwise comparisons.

We then show the $\Omega(k)$ lower bound. We simplify the problem and assume that the learner has already found $r_1, r_2, ..., r_k, r_{k+1}$ and the remaining goal is to identify $r_{k+1}$. Here, for any $i$, $r_i$ is the unknown $i$-th best item. For each $m$-wise comparison, we can exclude one item, i.e., the winner, from our consideration. Thus, we need $k$ comparisons to exclude all of $r_1, r_2, ..., r_k$, which implies the $\Omega(k)$ sample complexity lower bound. The proof is complete by summing up the above two lower bounds. □

### A.2  Proof of Theorem 2

**Theorem 2.** *MQSelect returns the top-$k$ items of $S$ after $O(\frac{n}{m} + k)$ comparisons in expectation.*

*Proof.* In the algorithm, we randomly choose $(m-1)$ items as the pivots. These pivots separate the other items into $m$ piles according to the ranks of these items. These piles are denoted as $A_1, A_2, ..., A_m$. Let $\tau$ be the first iteration $t$ such that $|D_t| \geq k$, i.e., the algorithm either terminates or enters the call of QS at iteration $\tau$. When the algorithm returns at the last line, we let $\tau = m$. We recall $n = |S|$ is the total number of items.

The key of the proof is to upper bound the size of $A_\tau$, which is stated in Lemma 15. The proof of Lemma 15 is provided in Section A.13 of the supplementary material.

**Lemma 15.** $\mathbb{E}[|A_\tau|] \leq 8 + \frac{2n}{m}$.

For each iteration $t$, MQSelect finds one pivot $v_t$, and thus, the number of comparisons used for finding pivots is at most $\tau$. Let $i$ be an item in $A_t$. When $t \leq \min\{\tau, m-1\}$, item $i$ wins a comparison that involves pivot $v_t$ and also loses one comparison with each of the pivots $v_1, v_2, ..., v_{t-1}$. When $t = m$, item $i$ loses $m - 1$ comparisons, each with a pivot, and wins no comparison with the pivots. For any time that a pivot wins a comparison, $m - 1$ items will each lose a comparison if we ignore those comparisons over sets with sizes smaller than $m$. When taking these sets with smaller sets into consideration, the expected number of items compared in these sets is at least $\frac{m+1}{2}$, which implies at least $\frac{m-1}{2}$ losers in expectation for each comparison. Thus, the number of comparisons required to identify $A_1, A_2, ..., A_\tau$ is at most

$$\sum_{t=1}^{\tau} \left[ |A_t| \cdot \left( 1 + \frac{2(t-1)}{m-1} \right) \right] \leq 5 \sum_{t=1}^{\tau} |A_t|.$$

Therefore, except the call of QS (if it exists), MQSelect conducts at most $\tau + 5 \sum_{t=1}^{\tau} |A_t|$ comparisons. This implies

$$T(S, m, k) \leq \tau + 5 \sum_{t=1}^{\tau} |A_t| + T(A_\tau, k),$$

where $T(A_\tau, k)$ is the number of comparisons QS needs for top-$k$ selection from $A_\tau$. We have $T(A_\tau, k) = O(|A_\tau|)$ and $\tau \leq k$. Also, since MQSelect returns at the first iteration $t$ where $|D_t| \geq k$,

we have

$$\sum_{t=1}^{\tau} |A_t| \leq |D_\tau| \leq k + |A_\tau|.$$

Therefore, we have

$$T(S, m, k) \leq k + 5(k + |A_\tau|) + O(|A_\tau|) = O(k + |A_\tau|),$$

which along with Lemma 15 ($\mathbb{E}[|A_\tau|] \leq 8 + \frac{2n}{m}$) completes the proof. $\qquad \square$

### A.3 Proof of Proposition 3

**Proposition 3.** *There is an $n$-sized instance such that to find the top-$k$ items with confidence $1 - \delta$ by using $m$-wise non-deterministic winner feedback, any algorithm needs at least $\Omega(k + \frac{n}{m} \log \frac{k}{\delta})$ comparisons in expectation.*

*Proof.* Towards contradiction, we assume that there is an algorithm $\mathcal{A}$ that can find the top-$k$ items of $[n]$ by using $o(\frac{n}{m} \log \frac{k}{\delta})$ $m$-wise comparisons in expectation. In the proof, we assume that every comparison returns the correct result with probability $\frac{2}{3}$, and will not explicitly state this for simplicity.

We recall that the authors of [9, 17] proved that there is an algorithm that can find the best item among $m$ items with confidence $\frac{2}{3}$ by using $O(m)$ pairwise comparisons. Thus, for any set with size $m$, we can use $O(m)$ pairwise comparisons with error probability $\frac{1}{3}$ to simulate an $m$-wise comparison with error probability at most $\frac{1}{3}$. According to our assumption, since algorithm $\mathcal{A}$ can find the top-$k$ items of $S$ with confidence $1 - \delta$ by $o(\frac{n}{m} \log \frac{k}{\delta})$ $m$-wise comparisons, by simulating the $m$-wise comparisons by pairwise comparisons, we can construct an algorithm which finds the top-$k$ items of $S$ with confidence $1 - \delta$ by $o(n \log \frac{k}{\delta})$ pairwise comparisons. However, the work in [9] proved that top-$k$ selection with confidence $1 - \delta$ requires at least $\Omega(n \log \frac{k}{\delta})$ pairwise comparisons, leading to a contradiction. Thus, such algorithm $\mathcal{A}$ does not exist, and the $\Omega(\frac{n}{m} \log \frac{k}{\delta})$ lower bound follows.

The $\Omega(k)$ lower bound follows from the lower bound for top-$k$ selection from deterministic comparisons stated in Theorem 1. Combining these two lower bounds, we get the desired lower bound and complete the proof of Proposition 3. $\qquad \square$

### A.4 Proof of Lemma 4

**Lemma 4.** *BKS terminates after $O((\frac{n}{m} + k) \log \frac{nk \log m}{m\delta})$ comparisons in expectation, and with probability at least $1 - \delta$, returns the top-$k$ items of $S$.*

*Proof.* Similar to the proof of Theorem 1, we let $\tau$ be the first iteration $t$ such that $|D_t| \geq k$, i.e., the algorithm either terminates or enters the call of QS at iteration $\tau$. When the algorithm returns at the last line, we let $\tau = m$. We recall $n = |S|$ is the total number of items. To prove the lemma, we need to show that in the execution of BKS, $A_\tau$ is of size at most $T_1 = O(\frac{n}{m} \log \frac{m}{\delta})$ with a probability at least $1 - \delta_0$, where $\delta_0 := \frac{\delta}{3}$.

We let $X_1, X_2, ..., X_{m-1}$, $L$, and $R$ denote the same things as in the proof of Lemma 15 (See Section A.13). We note that $|A_\tau| \leq L + R$. In the proof of Lemma 15, it has been shown in Eq (2) that

$$\mathbb{P}\{L + R = s\} \leq \frac{(m-1)(m-2)(s+1)}{n(n-1)} \cdot \left(1 - \frac{s-1}{n-2}\right)^{m-3}.$$

We define $f(s) := \frac{(m-1)(m-2)(s+1)}{n(n-1)} \cdot (1 - \frac{s-1}{n-2})^{m-3}$. When $s \geq \frac{n-m+2}{m-2}$, we have

$$f'(s) = \frac{(m-1)(m-2)}{n(n-1)} \left[\left(1 - \frac{s-1}{n-2}\right)^{m-3} - \frac{(m-3)(s+1)}{n-2}\left(1 - \frac{s-1}{n-2}\right)^{m-4}\right] \leq 0,$$

and thus, for $s \geq \frac{n-m+2}{m-2}$, $f(s)$ is non-increasing.

Now let $s \geq \frac{n-m+2}{m-2}$ be given, and since the maximal of $L + R$ is $n - m + 2$, we have

$$\mathbb{P}\{L + R > s\} \leq (n - m + 2 - s) \cdot f(s)$$
$$= (n - m + 2 - s) \cdot \frac{(m-1)(m-2)(s+1)}{n(n-1)} \cdot \left(1 - \frac{s-1}{n-2}\right)^{m-3}$$
$$\leq (n-2)\left(1 - \frac{s}{n-2}\right) \cdot \frac{(m-1)(m-2)(s+1)}{n(n-1)} \cdot \left(1 - \frac{s-1}{n-2}\right)^{m-3}$$
$$= \frac{(n-2)^2(m-1)(m-2)}{n(n-1)} \cdot \frac{s+1}{n-2} \cdot \left(1 - \frac{s-1}{n-2}\right)^{m-3} \cdot \left(1 - \frac{s}{n-2}\right)$$
$$\leq \frac{(n-2)^2(m-1)(m-2)}{n(n-1)} \cdot \frac{2(s-1)}{n-2} \cdot \left(1 - \frac{s-1}{n-2}\right)^{m-2}$$
$$\leq 2(m-1)(m-2)x(1-x)^{m-2},$$

where $x := \frac{s-1}{n-2}$. When $x \geq \frac{1}{m-2} \log \frac{2(m-1)(m-2)}{\delta_0}$, we have

$$\log(2(m-1)(m-2)x(1-x)^{m-2}) = \log(2(m-1)(m-2)) + \log x + (m-2)\log(1-x)$$
$$\leq \log(2(m-1)(m-2)) + 0 - (m-2)x$$
$$\leq \log(2(m-1)(m-2)) - \log \frac{2(m-1)(m-2)}{\delta_0} = \log \delta_0,$$

and thus, $2(m-1)(m-2)x(1-x)^{m-2} \leq \delta_0$. By $x := \frac{s-1}{n-2}$, we conclude that

$$s \geq 1 + \frac{n}{m-2} \log \frac{2(m-1)(m-2)}{\delta_0} \implies \mathbb{P}\{L + R > s\} \leq \delta_0.$$

Since $|A_\tau| \leq L + R$, we have that

$$|A_\tau| \leq 1 + \frac{n}{m-2} \log \frac{2(m-1)(m-2)}{\delta_0} \quad \text{with probability at least } 1 - \delta_0.$$

We also recall that $T_1 = 1 + \frac{n}{m-2} \log \frac{2(m-1)(m-2)}{\delta_0}$.

**Correctness.** We let $\mathcal{E}$ be the event that $|A_\tau| \leq T_1$. We have $\mathbb{P}\{\mathcal{E}\} \leq \delta_0$. In the proof of the correctness, we assume that $\mathcal{E}$ happens. Except the call of QS, MQSelect conducts at most $(6k + 5|A_\tau|)$ comparisons. Since each comparison is replaced by a call of BC with confidence $1 - \frac{\delta_0}{6k+5T_1} \geq 1 - \frac{\delta_0}{6k+5|A_\tau|}$, by the union bound, with probability at least $1 - \delta_0$, all these calls of BC return correct results. Finally, since the call of QS uses at most $|A_\tau|^2$ comparisons and each comparison is replaced by a call of BC with confidence $1 - \frac{\delta_0}{|A_\tau|^2}$, by the union bound, QS returns the correct result with probability at least $1 - \delta_0$. Therefore, BKS returns the top-$k$ items of $S$ with probability at least $1 - 3\delta_0 = 1 - \delta$. This proves the correctness.

**Sample complexity.** By Theorem 2, MQSelect conducts $O(\frac{n}{m} + k)$ comparisons in expectation except the call of QS. In BKS, each comparison is replaced by a call of BC with confidence $1 - \frac{\delta_0}{6k+5T_1}$, which conducts $O(\log \frac{6k+5T_1}{\delta_0}) = O(\log \frac{nk \log m}{m\delta_0})$ comparisons. Thus for BKS, Line 2 conducts $O((\frac{n}{m} + k) \log \frac{nk \log m}{m\delta_0})$ comparisons in expectation.

For the call of QS, its expected sample complexity is $\mathbb{E}[O(|A_\tau|)]$. Each comparison of QS is replaced by a call of BC with confidence $1 - \frac{\delta_0}{|A_\tau|^2}$, and thus, Line 3 conducts $O(|A_\tau| \log \frac{|A_\tau|}{\delta_0})$ comparisons in expectation. Here, we show Lemma 16 for upper bounding $\mathbb{E}[|A_\tau| \log |A_\tau|]$.

**Lemma 16.** $\mathbb{E}[|A_\tau| \log |A_\tau|] = O(\frac{n}{m} \log \frac{n}{m})$.

Lemma 16 implies that $O(\mathbb{E}[|A_\tau| \log \frac{|A_\tau|}{\delta_0}]) = O(\frac{n}{m} \log \frac{n}{m\delta})$, and thus, Line 3 conducts at most $O(\frac{n}{m} \log \frac{n}{m\delta})$ comparisons in expectation. By $\delta_0 = \frac{\delta}{3}$, the expected sample complexity of BKS is $O((\frac{n}{m} + k) \log \frac{nk \log m}{m\delta})$. This proves the sample complexity, and the proof of Lemma 4 is complete. $\qquad \square$

## A.5  Proof of Theorem 5

**Theorem 5.** *MTKS terminates after $O((\frac{n}{m} + k) \log \frac{k \log m}{\delta})$ comparisons in expectation, and with probability at least $1 - \delta$, returns the top-$k$ items of $S$.*

*Proof.* **Correctness.** Let $t$ be given, $\delta_t = \frac{6\delta}{\pi^2 t^2}$, and $U_t$ be the set of top-$k$ items of $R_t$. Since $R_t = \bigcup_{i \in [d]} A_i$ and $A_i$'s are disjoint, for each item $u$ in $U_t$, there is a set $A_{s_u}$ that contains $u$. By Lemma 4, for any $u$ in $U_t$, the call of BKS on set $A_{s_u}$ returns its top-$k$ items with probability at least $1 - \frac{6\delta}{\pi^2 t^2 k} = 1 - \frac{\delta_t}{k}$, and thus, with probability at least $1 - \delta_t$, the calls of BKS on sets $(A_{s_u} : u \in U_t)$ all return their top-$k$ items. Since any $u$ in $U_t$ is one of the top-$k$ items of $R_t$, and thus, $u$ is also one of the top-$k$ items of $A_{s_u}$. Therefore, with probability at least $1 - \delta_t$, the top-$k$ items of $R_t$ are all added to the set $R_{t+1}$. By $\sum_{t=1}^{\infty} \delta_t = \frac{6\delta}{\pi^2} \sum_{t=1}^{\infty} \frac{1}{t^2} = \delta$, the correctness of MTKS follows.

**Sample complexity.** First, we consider the case where $n > mk$. Let $\tau$ be the number of rounds the algorithm performs before termination. Since at each time, we divide set $R_t$ into $\lceil \frac{|R_t|}{mk} \rceil$ sets, and for each set, only $k$ items are put to $R_{t+1}$. Thus, $|R_{t+1}| = k \lceil \frac{|R_t|}{mk} \rceil$, which implies $|R_t| \le \frac{c_1 n}{m^{t-1}}$, where $c_1$ is some positive constant. For each round $t$, there are at most $\lceil \frac{|R_t|}{mk} \rceil$ calls of BKS. Each call is on at most $mk$ items. By Lemma 4, each call conducts $O((\frac{mk}{m} + k) \log \frac{mk^2 \log m}{m \delta_t}) = O(k \log \frac{k \log m}{\delta_t})$ comparisons in expectation. Thus, the total expected sample complexity of MTKS is upper bounded by

$$O\Big( \sum_{t=1}^{\tau} \Big( \frac{n}{m^{t-1}} \cdot \frac{1}{mk} \cdot k \log \frac{k \log m}{\delta_t} \Big) \Big) = O\Big( \sum_{t=1}^{\tau} \Big( \frac{n}{m^t} \cdot \log \frac{kt \log m}{\delta} \Big) \Big)$$

$$= O\Big( \frac{n}{m} \log \frac{k \log m}{\delta} \Big).$$

Since $n > mk$, i..e, $\frac{n}{m} > k$, the upper bound can also be written as $O((\frac{n}{m} + k) \log \frac{k \log m}{\delta})$.

For the case where $n \le mk$, MTKS directly calls BKS, which yields a sample complexity $O((\frac{n}{m} + k) \log \frac{nk \log m}{m \delta})$. Since $\frac{n}{m} \le k$, the sample complexity reduces to $O((\frac{n}{m} + k) \log \frac{k \log m}{\delta})$. This completes the proof of the sample complexity, and the proof of Theorem 5 is complete. $\square$

## A.6  Proof of Theorem 7

**Theorem 7.** *There is an $n$-sized instance such that to get the full ranking with confidence $1 - \delta$ from $m$-wise winner feedback, any algorithm needs at least $\Omega(n \log \frac{n}{\delta})$ comparisons.*

*Proof.* Let $\mathcal{A}$ be an arbitrary algorithm for finding the full ranking by $m$-wise noisy comparisons under the winner feedback model. We assume that $n$ is even, and the case where $n$ is odd can be proved by ignoring an item. Let $s := \frac{n}{2}$. For a set $M$ and an item $i$ in $M$, we use $p_{i,M}$ to denote the probability that item $i$ wins the comparison over set $M$. We recall that the unknown true ranking is $r_1 \succ r_2 \succ \cdots \succ r_n$. Define $\Pi := \{0,1\}^s$, and for each $\pi = (\pi_1, \pi_2, ..., \pi_s)$ in $\Pi$, we define the following hypothesis.

**Hypothesis $\mathcal{H}_\pi$.** The true ranking of $[n]$ is $q_1 \succ q_2 \succ \cdots \succ q_n$ where for any $i$ in $[s]$, $(q_{2i-1}, q_{2i}) = (2i-1, 2i)$ if $\pi_i = 0$ and $(q_{2i-1}, q_{2i}) = (2i, 2i-1)$ otherwise.

We have the following further assumptions. For any value of $\pi$ and set $M$, if the best two items of $M$ are not $2i$ and $(2i-1)$ for any $i$ in $[s]$, then for all $j$ in $M$, $p_{j,M}$ is the same under all values of $\pi$; if the best two items of $M$ are $2i$ and $(2i-1)$ for some $i$ in $[s]$, then $(p_{2i-1,M}, p_{2i,M}) = (\frac{2}{3}, \frac{1}{3})$ if $\pi_i = 0$ and $(p_{2i-1,M}, p_{2i,M}) = (\frac{1}{3}, \frac{2}{3})$ if $\pi_i = 1$.

Now, we are interested in the following problem $\mathcal{P}_1$. We note that there is at most one hypothesis $\mathcal{H}_\pi$ holds and we denote it by $\mathcal{H}_{\pi^*}$.

**Problem $\mathcal{P}_1$.** Assume that there is one $\pi^*$ in $\Pi$ such that $\mathcal{H}_{\pi^*}$ is true, and $\mathbb{P}\{\pi^* = \pi\} = \frac{1}{2^s}$ for any $\pi$. We want to find the value of $\pi^*$ with confidence $1 - \delta$ by using $m$-wise comparisons.

First, by using $\mathcal{A}$ to find the true ranking of $[n]$, one can find the value of $\pi_i$ by checking the whether the $(2i-1)$-th item in the true ranking is $(2i-1)$ or $2i$ for any $i$. Thus, $\mathcal{A}$ solves the problem $\mathcal{P}_1$.

Second, we show the sample complexity lower bound of $\mathcal{P}_1$. For any set $M$ of which the best two items are not $2i$ or $(2i - 1)$, the values of $p_{2i,M}$ and $p_{2i-1,M}$ are the same under any value of $\pi_i$, which implies that the comparisons over set $M$ does not contribution any information towards the value of $\pi_i$. Thus, only the comparisons over the sets $M$ of which the best two items are $2i$ and $(2i - 1)$ can be used to recover the value of $\pi$.

Now let $\mathcal{M}_i$ be the collection of sets $M_i$ of which the best two items are $2i$ and $(2i - 1)$ for some $i$ in $[s]$. If $\pi_i^* = 0$, then $(p_{2i-1,M_i}, p_{2i,M_i}) = (\frac{2}{3}, \frac{1}{3})$; and if $\pi_i^* = 1$, then $(p_{2i-1,M_i}, p_{2i,M_i}) = (\frac{1}{3}, \frac{2}{3})$. The comparisons over the sets not in $\mathcal{M}_i$ do not provide any information about the value of $\pi_i^*$. Therefore, recovering the value of $\pi_i^*$ is the same as determining whether a Bernoulli distribution is of parameter $\frac{2}{3}$ or $\frac{1}{3}$. According to [17], to determine this parameter with confidence $\delta_i$, at least $\Omega(\log \frac{1}{\delta_i})$ samples are required in expectation, which implies that to recover the value of $\pi_i^*$ with confidence $1 - \delta_i$, at least $\Omega(\log \frac{1}{\delta_i})$ comparisons over sets in $\mathcal{M}_i$ are needed in expectation.

To get the value of $\pi^*$ with confidence $1 - \delta$, we need to find the value of $\pi_i^*$ with confidence $1 - \delta_i$ for any $i$ in $[s]$, where $\prod_{i \in [s]}(1 - \delta_i) \geq 1 - \delta$. Thus, to solve $\mathcal{P}_1$ with confidence $1 - \delta$, the expected number of comparisons required is at least

$$\Omega\Big( \min \Big\{ \sum_{i \in [s]} \log \frac{1}{\delta_i} : \prod_{i \in [s]} (1 - \delta_i) \geq 1 - \delta \Big\} \Big).$$

We note that the set $\{(\delta_1, \delta_1, ..., \delta_s) : \prod_{i \in [s]}(1 - \delta_i) \geq 1 - \delta\}$ is convex, and the function $\sum_{i \in [s]} \log \frac{1}{\delta_i}$ is convex with respect to $(\delta_1, \delta_1, ..., \delta_s)$. Thus, by Jasen's Inequality, we have

$$\sum_{i \in [s]} \log \frac{1}{\delta_i} \geq s \log \frac{s}{\delta} = \Omega\Big( n \log \frac{n}{\delta} \Big),$$

which implies that the sample complexity lower bound of the problem $\mathcal{P}_1$ is $\Omega(n \log \frac{n}{\delta})$. Since $\mathcal{A}$ solves $\mathcal{P}_1$, the sample complexity of $\mathcal{A}$ is also lower bounded by $\Omega(n \log \frac{n}{\delta})$. Algorithm $\mathcal{A}$ is arbitrary, and this completes the proof of Theorem 7. $\qquad\square$

## A.7 Proof of Theorem 8

**Theorem 8.** *To get the top-$k$ items of $n$ items from deterministic $m$-wise full-ranking feedback, $\Theta(\frac{n}{m})$ comparisons are necessary and sufficient.*

*Proof.* **Lower bound.** For the lower bound, we can see that to get the top-$k$ items, each item needs to be involved in at least one comparison, and thus, we need at least $\frac{n}{m}$ comparisons. The desired lower bound follows.

**Upper bound.** The $O(\frac{n}{m})$ sample complexity upper bound can be achieved by the algorithm described in Algorithm 6. We first present this algorithm and then prove the upper bound.

The proof of the upper bound of MQSelect-FRF follows the similar steps as that of Theorem 2. When $|S| \leq m$, the number of comparisons required is 1 and the upper bound follows. In the rest of the proof, we assume $|S| > m$.

From Line 4 to Line 15, MQSelect-FRF conducts at most $(1 + \lceil \frac{|S|-h}{m-h} \rceil) = O(\frac{n}{m})$ comparisons. Similar to MQSelect, MQSelect-FRF randomly chooses $h$ pivots and split the rest of the items to $(h + 1)$ piles according to there order relationship with the pivots. Define $\tau := \inf\{t : |D_t| \geq k\}$. If the algorithm returns at the last line, we let $\tau = h + 1$. With the same steps as in the proof of Lemma 15, we have $\mathbb{E}|A_\tau| \leq 8 + \frac{2n}{h+1}$. QS conducts $O(|A_\tau|) = O(\frac{n}{m})$ comparisons in expectation. Thus, the total number of comparisons conducted by MQSelect-FRF is $O(\frac{n}{m})$ in expectation. This completes the proof of Theorem 8. $\qquad\square$

## A.8 Proof of Proposition 9

**Proposition 9.** *There is an $n$-sized instance such that to find the top-$k$ items by $m$-wise non-deterministic full-ranking feedback, any algorithm needs $\Omega(\frac{n}{m} \log_m \frac{k}{\delta})$ comparisons.*

---
**Algorithm 6** Multi-wise Quick-Select from Full-Ranking Feedback$(S, m, k)$ (MQSelect-FRF)
---
1: **if** $|S| \leq m$ **then**
2:     Compare $S$ and return the top-$k$ items according to the returned full ranking;
3: **end if**
4: Randomly choose $h := \lfloor \frac{m}{2} \rfloor$ items and form pivot set $V$;
5: Compare $V$ to get its full ranking $v_1 \succ v_2 \succ \cdots \succ v_h$;
6: Define $v_{h+1}$ as an item that ranks lower than any other item; Define $v_0$ as an items that ranks
    higher than any other item;
7: $R \leftarrow S - V$; $A_i \leftarrow \emptyset$ for $i = 1, 2, ..., h+1$;
8: **while** $R \neq \emptyset$ **do**
9:     Choose $(m - h)$ items from $R$ and form set $E$;
10:     Compare set $E \cup V$ and get the ranking;
11:     **for** item $i$ in $E$ **do**
12:         Add item $i$ to set $A_j$ if $v_{j-1} \succ i \succ v_j$;
13:     **end for**
14:     $R \leftarrow R - E$;
15: **end while**
16: $D_0 \leftarrow \emptyset$;
17: **for** $t = 1, 2, ..., h$ **do**
18:     $D_t \leftarrow D_{t-1} \cup A_t \cup \{v_t\}$;
19:     **if** $|D_t| = k$ **then**
20:         **return** $D_t$;
21:     **else if** $|D_t| = k + 1$ **then**
22:         **return** $D_t - \{v_t\}$;
23:     **else if** $|D_t| > k + 1$ **then**
24:         **return** $D_{t-1} \cup \mathrm{QS}(A_t, k - |D_{t-1}|)$;
25:     **end if**
26: **end for**
27: **return** $D_h \cup \mathrm{QS}(A_{h+1}, k - |D_h|)$;
---

*Proof.* First, we note that by Theorem 12 of [17], to get the full ranking of $m$ items with confidence $\frac{2}{3}$ from non-deterministic pairwise comparisons, $O(m \log m)$ comparisons are sufficient. In other words, we can simulate an $m$-wise full-ranking-feedback comparison by $O(m \log m)$ pairwise comparisons. We then note that by Theorem 12 of [17], to get the top-$k$ items of $[n]$ with confidence $1 - \delta$ from pairwise non-deterministic comparisons, at least $\Omega(n \log \frac{k}{\delta})$ comparisons are needed. Thus, if there is an algorithm $\mathcal{A}$ that can find the top-$k$ items with confidence $1 - \delta$ by $o(\frac{n}{m} \log_m \frac{k}{\delta})$ comparisons, then by using pairwise comparisons to simulate $m$-wise comparisons, we can find the top-$k$ items with confidence $1 - \delta$ by $o(n \log \frac{k}{\delta})$ pairwise comparisons. This contradicts the lower bound stated in Theorem 12 of [17]. Therefore, such algorithm $\mathcal{A}$ does not exist and the desired lower bound follows. This completes the proof of Proposition 9. □

### A.9 Proof of Theorem 10

**Theorem 10.** *MTKS-FRF terminates after $O(\frac{n}{m} \log \frac{\min\{n/m, k\} \log m}{\delta})$ comparisons in expectation, and with probability at least $1 - \delta$, returns the top-$k$ items of $S$.*

*Proof.* We first introduce the subroutine BKS-FRF (Basic $k$-selection from Full-Ranking Feedback), which is described in Algorithm 7. Its theoretical performance is formally stated in Lemma 17.

**Lemma 17.** *BKS-FRF terminates after $O(\frac{n}{m} \log \frac{n \log m}{m \delta})$ comparisons in expectation, and with probability at least $1 - \delta$, returns the top-$k$ items of $S$.*

Then, similar to MTKS, we develop a tournament-like algorithm MTKS-FRF (Multi-wise Tournament $k$-Selection from Full-Ranking Feedback), which is described in Algorithm 8.

**Correctness.** Let $t$ be given, $\delta_t = \frac{6\delta}{\pi^2 t^2}$, and $U_t$ be the set of top-$k$ items of $R_t$. Since $R_t = \bigcup_{i \in [d]} A_i$ and $A_i$'s are disjoint, for each item $u$ in $U_t$, there is a set $A_{s_u}$ that contains $u$. By Lemma 17,

---

**Algorithm 7** BKS from Full-Ranking Feedback$(S, m, k, \delta)$ (BKS-FRF)

---

1: $h \leftarrow \lfloor \frac{m}{2} \rfloor$; $\delta_0 \leftarrow \frac{\delta}{3}$; $T_2 \leftarrow 1 + \frac{n}{h-1} \log \frac{2h(h-1)}{\delta}$; $\delta_1 \leftarrow \frac{\delta_0}{1 + \frac{n-h}{m-h} + T_2}$ if $m > 2$; $\delta_1 \leftarrow \frac{\delta_0}{|S|^2}$ if $m = 2$;

2: Run MQSelect-FRF on $M$, but using calls of BC with confidence $1 - \delta_1$ to replace multi-wise comparisons (except those in the call of QS);

3: For the call of QS, we replace the pairwise comparisons with calls of BC with confidence $1 - \frac{\delta_0}{|A_t|^2}$;

---

**Algorithm 8** MTKS from Full-Ranking Feedback$(S, m, k, \delta)$ (MTKS-FRF)

---

1: **if** $|R_t| \leq mk$ **then**
2:     **return** BKS-FRF$(S, m, k, \delta)$;
3: **end if**
4: Set $t \leftarrow 0$ and $R_1 \leftarrow S$;
5: **repeat**
6:     $t \leftarrow t + 1$; $\delta_t \leftarrow \frac{6\delta}{\pi^2 t^2}$;
7:     Distribute $R_t$ to $\lceil \frac{|R_t|}{mk} \rceil$ disjoint sets $A_1, A_2, ..., A_d$, each with size at most $mk$;
8:     For $i \in [d]$, let $T_i \leftarrow$ BKS-FRF$(A_i, m, k, \frac{\delta_t}{k})$;
9:     $R_{t+1} \leftarrow \bigcup_{i \in [d]} A_i$;
10: **until** $|R_{t+1}| = k$
11: **return** $R_{t+1}$;

---

for any $u$ in $U_t$, the call of BKS-FRF on set $A_{s_u}$ returns its top-$k$ items with probability at least $1 - \frac{6\delta}{\pi^2 t^2 k} = 1 - \frac{\delta_t}{k}$, and thus, with probability at least $1 - \delta_t$, the calls of BKS-FRF on sets $(A_{s_u} : u \in U_t)$ all return their top-$k$ items. Since any $u$ in $U_t$ is one of the top-$k$ items of $R_t$, and thus, $u$ is also one of the top-$k$ items of $A_{s_u}$. Therefore, with probability at least $1 - \delta_t$, the top-$k$ items of $R_t$ are all added to the set $R_{t+1}$. By $\sum_{t=1}^{\infty} \delta_t = \frac{6\delta}{\pi^2} \sum_{t=1}^{\infty} \frac{1}{t^2} = \delta$, the correctness of MTKS-FRF follows.

**Sample complexity.** First, we consider the case where $n > mk$. Let $\tau$ be the number of rounds the algorithm performs before termination. Since at each time, we divide set $R_t$ into $\lceil \frac{|R_t|}{mk} \rceil$ sets, and for each set, only $k$ items are put to $R_{t+1}$. Thus, $|R_{t+1}| = k \lceil \frac{|R_t|}{mk} \rceil$, which implies $|R_t| \leq \frac{c_1 n}{m^{t-1}}$, where $c_1$ is some positive constant. For each round $t$, there are at most $\lceil \frac{|R_t|}{mk} \rceil$ calls of BKS-FRF. Each call is on at most $mk$ items, and by Lemma 17, each call conducts $O(\frac{mk}{m} \log \frac{mk \log m}{m\delta_t}) = O(k \log \frac{k \log m}{\delta_t})$ comparisons in expectation. Thus, the total sample complexity of MTKS is upper bounded by

$$O\Big( \sum_{t=1}^{\tau} \Big( \frac{n}{m^{t-1}} \cdot \frac{1}{mk} \cdot k \log \frac{k \log m}{\delta_t} \Big) \Big) = O\Big( \sum_{t=1}^{\tau} \Big( \frac{n}{m^t} \cdot \log \frac{kt \log m}{\delta} \Big) \Big)$$

$$= O\Big( \frac{n}{m} \log \frac{k \log m}{\delta} \Big).$$

For the case where $n \leq mk$, MTKS directly calls BKS-FRF, which yields a sample complexity $O(\frac{n}{m} \log \frac{n \log m}{m\delta})$. This completes the proof of the sample complexity, and the proof of Theorem 10 is complete. □

### A.10 Proof of Theorem 11

**Theorem 11.** *To get the full ranking of $n$ items by using $m$-wise comparisons under the full-ranking feedback model, any algorithm needs at least $\Omega(\frac{n \log n}{m \log m})$ comparisons in expectation.*

*Proof.* Let $\mathcal{A}$ be an arbitrary deterministic algorithm. Let $R$ be the permutation representing the true ranking. Since there are $n$ items, and each permutation has the same probability to be the true ranking, we have $H(R) = \log(n!)$, where $H(\cdot)$ is the information entropy.

Let $T$ be the number of comparisons conducted by $\mathcal{A}$. Let $M_t := \{i_{t,1}, i_{t,2}, ..., i_{t,m}\}$ be the $t$-th compared set and $\mathcal{M} := (M_1, M_2, ..., M_T)$ be the collection of compared sets. For the $t$-th comparison on $M_t$, if the returned true ranking is $(i_{t,s_1}, i_{t,s_2}, ..., i_{t,s_m})$, then we let $X_t = (s_1, s_2, ..., s_m)$, and we use $\mathcal{X} = (X_1, X_2, ..., X_T)$ to denote the collection of comparison results. Define $S_t := (M_t, X_t, \mathbb{1}_{t \leq T})$ as the state of the $t$-th comparison and $U_t := (S_1, S_2, ..., S_t)$ as the history till the $t$-th comparison.

Since the algorithm is deterministic, $M_1$ is deterministic, $M_t$ is determined by $U_{t-1}$, and $\mathbb{1}_{t \leq T}$ is determined by $(U_{t-1}, M_t, X_t)$. Thus, we have

$$
\begin{aligned}
H(U_t \mid U_{t-1}) =& H(S_t \mid U_{t-1}) \\
\leq& H(M_t \mid U_{t-1}) + H(X_t \mid M_t, U_{t-1}) + H(\mathbb{1}_{t \leq T} \mid M_t, X_t, U_{t-1}) \\
\leq& 0 + H(X_t \mid M_t) + 0 \\
=& \log(|M_t|!) \\
=& \log(m!).
\end{aligned}
$$

where $H(\cdot \mid \cdot)$ is the conditional information entropy. Then, we have

$$
\begin{aligned}
H(U_t) =& H(U_t \mid U_{t-1}) + H(U_{t-1} \mid U_{t-2}) + \cdots + H(U_1) \\
\leq& \log(m!) + \log(m!) + \cdots + \log(m!) \\
=& t \log(m!).
\end{aligned}
$$

Since the comparisons are deterministic, when the true ranking $R$ is given, the values of all $U_t$'s are deterministic, i.e., $H(U_t \mid R) = 0$. Therefore, we have

$$
\begin{aligned}
H(R \mid U_t) =& H(R) - I(R; U_t) \\
=& H(R) - (H(U_t) - H(U_t \mid R)) \\
\geq& \log(n!) - t \log(m!) + 0,
\end{aligned}
$$

where $I(\cdot; \cdot)$ is the mutual information between two random vectors.

We then invoke Fano's Inequality, which is stated in Fact 18.

**Fact 18** (Fano's Inequality [8])**.** *To recover the value of $X$ from $Y$ with error probability no more than $\delta$, it must hold that*

$$
H(X|Y) \leq H(\delta) + \delta \log(N - 1),
$$

*where $N$ is the number of values $X$ can take and $H(\delta) = \delta \log \frac{1}{\delta} + (1 - \delta) \log \frac{1}{1-\delta}$.*

Thus, to recover the true ranking (i.e., the value of $R$) with probability at least $\frac{1}{4}$ within $t$ comparisons, by Fano's Inequality, $\log(m!) = \Theta(m \log m)$, and $\log(n!) = \Theta(n \log n)$, it is required that

$$
\begin{aligned}
&H(R \mid U_t) \leq H(1/4) + \delta \log(n! - 1) \\
\implies& \log(n!) - t \log(m!) + 0 \leq \Theta(1) + \frac{1}{4} \log(n! - 1) \\
\implies& t \geq \frac{\log(n!) - \frac{1}{4} \log(n! - 1) - \Theta(1)}{\log(m!)} \\
\implies& t = \Omega\Big(\frac{n \log n}{m \log m}\Big).
\end{aligned}
$$

Any random algorithm cannot outperform the best deterministic algorithm, and thus, the desired lower bound also holds for random algorithms. This completes the proof of Theorem 11. $\qquad\square$

## A.11 Proof of Theorem 12

**Theorem 12.** *MQSort terminates after $O(\frac{n \log n}{m \log m})$ m-wise comparisons in expectation and returns the full ranking of $S$.*

*Proof.* Let set $S$ and $n = |S|$ be given. We say that the call of MQSort on $S$ is executed in round 1. In Line 16, it makes multiple recursive calls of MQSort and we say these calls are executed in round 2. Similarly, a call of MQSort executed in round $t$ may also make recursive calls and we say

these recursive calls are executed in round $(t + 1)$. We observe that during the execution of a call on $n'$ items, splitting the items to sets $A_i$'s takes at most $1 + \lceil \frac{|S-V|}{m-h} \rceil = O(\frac{n'}{m})$ comparisons. Since in each round $t$, the total number of items processed by all the recursive calls is at most $n$, the splitting processes in all these recursive calls use at most $O(\frac{n}{m})$ comparisons in total. Thus, the rest of the proof is to prove that the recursive calls on terminate after $O(\log_m n)$ rounds in expectation.

**Step 1.** Before finding the number of rounds of MQSort, we focus on a similar problem. We say that there are $N$ balls in total where $N$ is large enough. Then at round 1 we randomly put these $N$ balls into $M$ boxes where $M$ is large enough and $M \leq N$. In round 2, for each box, we randomly put the balls inside it into $M$ other boxes, i.e., the balls are now randomly distributed to $M^2$ boxes. For any round $t$ where $t > 2$, we do the same thing, i.e., randomly put the balls of each box into $M$ other boxes. We repeat this process until each box contains at most one ball. Let $T$ be the number of rounds before this process terminates. Our goal is to find an upper bound of $\mathbb{E}[T]$.

Now let $t \geq \log_M N$ be given, $i$ be a random ball, and $j$ be a random box that is introduced to this process in round $t$. We have that ball $i$ is in box $j$ with probability $\frac{1}{M^t}$. The places of these balls are independent, and thus, we have

$$\mathbb{P}\{T \leq t\} = \frac{M^t}{M^t} \cdot \frac{M^t - 1}{M^t} \cdot \frac{M^t - 2}{M^t} \cdots \frac{M^t - N + 1}{M^t} \geq \left( \frac{M^t - N + 1}{M^t} \right)^N.$$

Now we let $\delta \in (0, 1)$ be given. We define

$$\tau_\delta := 1 + \log_M \frac{N - 1}{1 - (1 - \delta)^{1/N}},$$

we have

$$\mathbb{P}\{T \leq \tau_\delta\} \geq \left( \frac{M^{\tau_\delta} - N + 1}{M^{\tau_\delta}} \right)^N = \left( 1 - \frac{N - 1}{M^{\tau_\delta}} \right)^N = \left( 1 - \frac{N - 1}{M^{\log_M \frac{N-1}{1-(1-\delta)^{1/N}}}} \right)^N = 1 - \delta.$$

Also, since $1 - (1 - \delta)^{1/N} \geq 1 - e^{-\delta/N} = \Omega(\frac{\delta}{N})$, we have

$$\tau_\delta = \log_M \left( \frac{N - 1}{1 - (1 - \delta)^{1/N}} \right) = O\left( \log_M \frac{N}{\delta} \right)$$

Therefore, by $\tau_1 = 1 + \log_M(N - 1)$, we conclude that

$$\begin{aligned}
\mathbb{E}[T] &= \sum_{t=0}^{\infty} \mathbb{P}\{T > t\} \\
&\leq \sum_{t > \tau_1} \mathbb{P}\{T > t\} + \tau_1 \\
&\leq \sum_{s=0}^{\infty} [(\tau_{2^{-s-1}} - \tau_{2^{-s}}) \cdot \mathbb{P}\{t > \tau_{2^{-s}}\}] + O(\log_M N) \\
&\leq \sum_{s=0}^{\infty} [\tau_{2^{-s-1}} \cdot \mathbb{P}\{t \geq \tau_{2^{-s}}\}] + O(\log_M N) \\
&= O\left( \sum_{s=1}^{\infty} \log_M \frac{N}{2^{-s-1}} \cdot 2^{-s} \right) + O(\log_M N) \\
&= O(\log_M N).
\end{aligned}$$

**Step 2.** We switch back to MQSort. We define $h = \lfloor \frac{m}{2} \rfloor$ and there are $h$ pivots among $n$ items that separate the set $S$ to $(h + 1)$ sets $A_0, A_1, ..., A_h$. For any integer $i$, we define $[i]_n := i \mod n$, i.e., the reminder of $i$ divided by $n$. We introduce a dummy item $v_0 := 0$ that ranks higher than any other item. Let $X$ and $Y$ be two random and distinct items in $\{v_0\} \cup S$ and we assume that the algorithm has finished Line 15, i.e., all the non-pivot items have been added to one of $A_0, A_1, ..., A_h$. We further define $B_j := A_j \cup \{v_j\}$ for any $j$. In the following analysis, we view pivots $v_1, v_2, ..., v_h$ as random distinct items selected from $S$. First we have

$$\mathbb{P}\{X \in A_0\} = \mathbb{P}\{\forall i > 0, v_i > X \geq 0\} = \mathbb{P}\{\forall i \neq 0, [v_i - v_0]_n > [X - v_0]_n\}.$$

For any set $A_j$ where $j > 0$, we have

$$\mathbb{P}\{X \in A_j\} = \mathbb{P}\{\forall i \neq j, [v_i - v_j]_n > [X - v_j]_n\}.$$

Therefore, when we change the set from $A_i$ to $A_j$, actually we are simply rotationally shifting the items with $[v_i - v_j]_n$ steps, which implies that $\mathbb{P}\{X \in A_0\} = \mathbb{P}\{X \in A_j\}$ for any $j$. Thus, for any $j$, we have

$$\mathbb{P}\{X \in A_j\} = \frac{1}{h} \text{ and } \mathbb{P}\{X \in A_j | X \neq v_j\} = \frac{1}{h}.$$

Now we assume that $X$ is not a pivot and $X$ is in the set $A_j$. In this case, we can define $S' = S - \{X\}$ and subtract the indexes of all items $i > X$ by one, and then the probabilities whether $Y$ is in $A_0$, $A_1$,..., or $A_h$ remain the same, i.e., for any $j$ and $l$, we have

$$\mathbb{P}\{Y \in A_j \mid X \in A_l, X \neq v_l\} = \frac{1}{h} \text{ and } \mathbb{P}\{Y \in A_j \mid X \in A_l, X \neq v_l, Y \neq V_j\} = \frac{1}{h}.$$

Repeating the above analysis. we have that with $s$ non-pivot items given, the probabilities whether the $(s + 1)$-th item is in some set $A_j$ remain $\frac{1}{h}$ for $s + 1 \leq n - h$.

**Step 3.** Therefore, with the above findings, we can view the distribution of non-pivot items as the "put balls into boxes" problem, where the non-pivot items are the balls and the sets $A_0, A_1, ..., A_h$ are the boxes. Since after each round, there are $h$ balls removed from the following splitting and the other conditions remain the same, let round $T'$ be the first round when each box has at most one ball, we have

$$\mathbb{E}[T'] = O(\log_{h+1} n) = O(\log_{\lceil \frac{m}{2} \rceil + 1} n) = O(\log_m n).$$

When each box has at most one ball, i.e., all sets $A_0, A_1, ..., A_h$ in all recursive calls of MQSort in round $T'$ have at most one item, the outer-most call will return, and the algorithm will terminate. Thus, there are $O(\log_m n)$ rounds in expectation. Recalling the fact that each round conducts at most at most $O(\frac{n}{m})$ comparisons, the algorithm returns after $O(\frac{n \log n}{m \log m})$ comparisons in expectation. This completes the proof of Theorem 12. $\qquad\square$

### A.12 Proof of Theorem 14

**Theorem 14.** *There is an algorithm that finds the full ranking of $n$ items by $m$-wise full-ranking feedback with confidence $1 - \delta$ and conducts $O(\frac{n}{m} \log \frac{n}{m\delta})$ comparisons in expectation.*

*Proof.* To get the desired upper bound, we need to invoke the full ranking algorithm in [9], which is denoted by $\mathcal{A}$. When all comparisons return correct results with probability $\frac{2}{3}$ and $m = 2$, $\mathcal{A}$ finds the true ranking of $n$ items with confidence $1 - \delta$ by using $O(n \log \frac{n}{\delta})$ pairwise comparisons. The algorithm is a variant of the insertion sorting, i.e., given a list of $n_1$ sorted items, it inserts a new item into this list. The algorithm repeats the insertion until all items have been inserted and the full ranking is found. In $\mathcal{A}$, inserting one item $i$ into a list of $n_i$ sorted items with confidence $1 - \delta_i$ uses $O(\log \frac{n_i}{\delta_i}) = O(\log \frac{n}{\delta_i})$ comparisons.

Now, we make the following modifications on $\mathcal{A}$. First, we randomly choose $m$ items and compare these $m$ items for $\Theta(\log \frac{n}{m\delta})$ times to get the ranking of these $m$ items with confidence $1 - \frac{m\delta}{4n}$. We use $R$ to denote this list. At each round, we choose $\frac{m}{2}$ new items (assuming that $m$ is even, and when $m$ is odd we can prove the upper bound similarly) and we want to insert them into the sorted list $R$. Since an $m$-wise comparison under the full-ranking feedback model can be viewed as doing $\frac{m}{2}$ pairwise comparisons at the same time, we do the insertion process for these $\frac{m}{2}$ new items simultaneously. During the insertion, we set the confidence at $1 - \frac{m\delta}{4n}$ and the depth of the insertion tree at $\Theta(\log \frac{n}{m\delta})$. After at most $O(\log \frac{n}{m\delta})$ comparisons, if at least $\frac{2}{3}$ proportion of the comparisons return correct results (which happens with probability at least $1 - \frac{m\delta}{4n}$), then all the $\frac{m}{2}$ new items will be inserted to correct places. We then compare these $\frac{m}{2}$ new items for $\Theta(\log \frac{n}{m\delta})$ times to get the ranking of them with confidence at least $1 - \frac{m\delta}{4n}$.

For each round of insertion, we use $O(\log \frac{n}{m\delta})$ comparisons, and each round inserts $\frac{m}{2}$ items. Thus, the insertion takes $O(\frac{n}{m} \log \frac{n}{m\delta})$ comparisons in total. We do $\Theta(\log \frac{n}{m\delta})$ comparisons for ranking the first $m$ items, and the same amount in each round of insertion for ranking the inserted new

items, which takes $O(\frac{n}{m} \log \frac{n}{m\delta})$ comparisons in total for inserting all items. Thus, the total sample complexity of this modified algorithm is at most $O(\frac{n}{m} \log \frac{n}{m\delta})$. This proves the sample complexity.

The ranking of the first $m$ item is correct with probability at least $1 - \frac{m\delta}{4n}$. For each round of insertion, the ranking of the newly inserted items is correct with probability at least $1 - \frac{m\delta}{4n}$, and the insertion is correct with probability at least $1 - \frac{m\delta}{4n}$. There are at most $\frac{n-m}{m/2} = \frac{2n}{m} - 2$ rounds of insertion. By the union bound, the total error probability of the modified algorithm is $\frac{m\delta}{4n}(1 + \frac{4n}{m} - 4) \leq \delta$. This proves the correctness and the proof of Theorem 14 is complete. $\qquad\square$

## A.13   Proof of Lemma 15

**Lemma 15.** $\mathbb{E}[|A_\tau|] \leq 8 + \frac{2n}{m}$.

*Proof.* The $(m-1)$ pivots are randomly chosen, and we use $X_1, X_2, ..., X_{m-1}$ to denote their positions in the true ranking, i.e., $X_i = j$ means that $X_i$ ranks the $j$-th largest in the true ranking. For simplicity, we define $X_0 := \infty$ that ranks higher than any item and $X_m := -\infty$ that ranks lower than any item. If $X_{i-1} \succ r_k \succeq X_i$, then we have $\tau = i$ and $r_k$ is in the set $A_i$ or $r_k = X_i$.

We put these $n$ items on a circle, i.e., we assume that item $r_1$ is on the right of $r_n$, $r_n$ is on the left of $r_1$, and they are adjacent. For any integer $a$, we define $[a]_n := a \mod n$, where $a \mod n$ is the reminder of $a$ divided by $n$. We further define

$$L := [k - X_{\tau-1}]_n, \text{ and } R := [X_\tau - k]_n.$$

After putting the $n$ items on the a circle, we can view $L$ as the distance from $r_k$ to the closest pivot on the left, and $R$ as the distance from $r_k$ to the closes pivot on the right. We have $|A_\tau| \leq L + R$, and our goal is to bound $\mathbb{E}[L + R]$.

Let $1 \leq s \leq n - m + 2$ be given. We have

$$\mathbb{P}\{L + R = s\} = \sum_{i,j \in [m-1]: i \neq j} \sum_{r=0}^{s} \left[ \mathbb{P}\{[k - X_i]_n = r\} \mathbb{P}\{[X_j - k]_n = s - r, \} \right.$$
$$\left. \cdot \mathbb{P}\{\forall l \neq i, j : s - r < [X_l - k]_n < n - r\} \right]$$
$$= (m-1)(m-2) \cdot \frac{s+1}{n(n-1)} \cdot \frac{\binom{n-s-1}{m-3}}{\binom{n-2}{m-3}}$$
$$\leq (m-1)(m-2) \cdot \frac{s+1}{n(n-1)} \cdot \left( \frac{n-s-1}{n-2} \right)^{m-3}$$
$$= \frac{(m-1)(m-2)(s+1)}{n(n-1)} \cdot \left( 1 - \frac{s-1}{n-2} \right)^{m-3}. \qquad (2)$$

Thus, we have

$$\mathbb{E}[L + S] \leq \sum_{s=1}^{n-m+2} \left[ s \cdot \frac{(m-1)(m-2)(s+1)}{n(n-1)} \cdot \left( 1 - \frac{s-1}{n-2} \right)^{m-3} \right]$$
$$\leq (m-1)(m-2) \sum_{s=1}^{n-m+2} \left[ \left( \frac{s-1}{n-2} \right)^2 \cdot \left( 1 - \frac{s-1}{n-2} \right)^{m-3} \right]$$

Now, we let $x := \frac{s-1}{n-2}$. Since in $[0, 1]$, the function $f(x) := x^2 (1-x)^{m-3}$ is non-negative and first increasing then decreasing, we have the following inequality.

$$\mathbb{E}[L+S] \le (m-1)(m-2)\Big[2\sup_{x\in[0,1]} x^2(1-x)^{m-3} + (n-2)\int_0^1 x^2(1-x)^{m-3}\,\mathrm{d}x\Big]$$

$$=(m-1)(m-2)\Big[2x^2(1-x)^{m-3}\big|_{x=\frac{2}{m}}$$

$$+ (n-2)\cdot\frac{(x-1)(1-x)^{m-3}[(m^2-3m)x^2+2(m-2)x+2]}{m(m-1)(m-2)}\Big|_0^1\Big]$$

$$=(m-1)(m-2)\Big[\frac{8}{m^2}\Big(1-\frac{2}{m}\Big)^3 + (n-2)\cdot\frac{2}{m(m-1)(m-2)}\Big]$$

$$\le (m-1)(m-2)\Big[\frac{8}{(m-1)(m-2)}\cdot 1^2 + n\cdot\frac{0-(-2)}{m(m-1)(m-2)}\Big]$$

$$=8+\frac{2n}{m}.$$

Therefore, we have $\mathbb{E}[|A_\tau|] \le \mathbb{E}[L+R] \le 8+\frac{2n}{m}$. This completes the proof of Lemma 15. $\square$

### A.14 Proof of Lemma 16

**Lemma 16.** $\mathbb{E}[|A_\tau|\log|A_\tau|] = O(\frac{n}{m}\log\frac{n}{m})$.

*Proof.* We let $X_1, X_2, ..., X_{m-1}$, $L$, and $R$ denote the same things as in the proof of Lemma 15 (See Section A.13). We note that $|A_\tau| \le L+R$. Let $X = |A_\tau|$.

According to Eq (2), we have

$$\mathbb{P}\{L+R=s\} \le \frac{(m-1)(m-2)(s+1)}{n(n-1)}\cdot\Big(1-\frac{s-1}{n-2}\Big)^{m-3},$$

We further use $x$ to denote $\frac{s-1}{n-2}$. The above result, by $X \le L+R$, implies

$$E[X\log X] \le \frac{(m-1)(m-2)}{n(n-1)}\sum_{s=1}^{n-m+2}\Big[s(s+1)\Big(1-\frac{s-1}{n-2}\Big)^{m-3}\log s\Big]$$

$$\le \frac{(m-1)(m-2)}{n(n-1)}\sum_{s=1}^{n-m+2}\Big[2(s-1)^2\Big(1-\frac{s-1}{n-2}\Big)^{m-3}(1+\log(s-1))\Big]$$

$$\le \frac{(m-1)(m-2)}{n(n-1)}\sum_{s=1}^{n-2}\Big[2(n-2)^2 x^2(1-x)^{m-3}(1+\log((n-2)x))\Big].$$

Here, we define $f(x) := x^2(1-x)^{m-3}\log((n-2)x)$. We have

$$f'(x) = x(1-x)^{m-4}[(2-(m-1)x)\log((n-2)x)+1-x].$$

When $\frac{1}{n-2} < x \le 1$, $f'(x)$ is first positive and then negative, though the actual transiting point is unknown. We also have $f(0) = f(\frac{1}{n-2}) = 0$. Thus, we have

$$\sum_{s=1}^{n-m+2} f(x) \le (n-2)\int_0^1 f(x)\,\mathrm{d}x - 2\min_{0\le x\le\frac{1}{n-2}} f(x) + 2\max_{\frac{1}{n-2}\le x\le 1} f(x).$$

First, we have

$$\int_0^1 f(x)\,\mathrm{d}x = \frac{2\log(n-2)}{m^3-3m^2+2m} + \frac{3-2\sum_{i=1}^m\frac{1}{i}}{m^3-3m^2+2m}$$

$$\le \frac{2\log n - 2\log m + O(1)}{m^3-3m^2+2m}$$

$$= O\Big(\frac{\log\frac{n}{m}}{m^3}\Big).$$

Second, when $0 \leq x \leq \frac{1}{n-2}$, we have

$$f(x) \geq x^2 \log((n-2)x),$$

which takes the minimum when

$$(x^2 \log((n-2)x))' = 2x \log((n-2)x) + x = 0,$$

i.e., when $x = \frac{e^{-1/2}}{n-2}$. Thus, we have

$$\min_{0 \leq x \leq \frac{1}{n-2}} f(x) \geq -\frac{1}{2}\Big(\frac{e^{-1/2}}{n-2}\Big)^2 = -\Theta(n^{-2}).$$

Third, when $x \geq \frac{1}{n-2}$, we have $f(x) \leq x^2(1-x)^{m-3} \log n$. $x^2(1-x)^{m-3}$ is maximal when $(x^2(1-x)^{m-3})' = 0$, i.e., $x = \frac{2}{m-1}$. Thus,

$$\max_{\frac{1}{n-2} \leq x \leq 1} f(x) \leq \Big(\frac{2}{m-1}\Big)^2\Big(1 - \frac{2}{m-1}\Big)^{m-3} \log n = O\Big(\frac{\log n}{m^2}\Big).$$

Therefore, we have

$$\sum_{s=1}^{n-m+2} f(x) = (n-2) \cdot O\Big(\frac{\log n}{m^3}\Big) + O\Big(\frac{1}{n^2}\Big) + 2 \cdot O\Big(\frac{\log n}{m^2}\Big)$$

$$= O\Big(\frac{n \log \frac{n}{m}}{m^3}\Big).$$

Similarly, since for $0 \leq x \leq 1$, $x^2(1-x)^{m-3}$ first increases and then decreases, we have

$$\sum_{s=1}^{n-m+2} x^2(1-x)^{m-3} \leq (n-2) \int_0^1 x^2(1-x)^{m-3} \, \mathrm{d}x + 2 \max_{0 \leq x \leq 1} x^2(1-x)^{m-3}$$

$$= \frac{2(n-2)}{m^3 - 3m^2 + 2m} + 2\Big(\frac{2}{m-1}\Big)^2\Big(1 - \frac{2}{m-1}\Big)^{m-3}$$

$$= O\Big(\frac{n}{m^3}\Big).$$

With the above results, we have

$$\mathbb{E}[X \log X] = \frac{(m-1)(m-2)}{n(n-1)} \cdot 2(n-2)^2\Big(O\Big(\frac{n \log \frac{n}{m}}{m^3}\Big) + O\Big(\frac{n}{m^3}\Big)\Big)$$

$$= O\Big(\frac{n}{m} \log \frac{n}{m}\Big).$$

This completes the proof of Lemma 16. $\qquad\square$

### A.15 Proof of Lemma 17

**Lemma 17.** *BKS-FRF terminates after* $O(\frac{n}{m} \log \frac{n \log m}{m\delta})$ *comparisons in expectation, and with probability at least* $1 - \delta$, *returns the top-k items of* $S$.

*Proof.* If QS is called, then we use $A_\tau$ to denote the set $A_t$ inputed to QS. To prove the lemma, we need to show that in the execution of MQSelect-FRF, $A_\tau$ is of size at most $T_1 = O(\frac{n}{m} \log \frac{m}{\delta})$ with a probability at least $1 - \delta_0$, where $\delta_0 := \frac{\delta}{3}$.

We let $X_1, X_2, ..., X_{m-1}, L$, and $R$ denote the same things as in the proof of Lemma 15. By similar steps as in the proof of Lemma 4, we have that with probability at least $1 - \delta_0$, $|A_\tau| \leq T_2$. Here we note that the only that is changed is the number of pivots (i.e., from $(m-1)$ to $h$), and we only need to change $m$ to $(h+1)$ in the formula of $T_2$.

**Correctness.** We let $\mathcal{E}$ be the event that $|A_\tau| \leq T_2$. We have $\mathbb{P}\{\mathcal{E}\} \leq \delta_0$. In the proof of the correctness, we assume that $\mathcal{E}$ happens. Except the call of QS, MQSelect conducts at most

$(1 + \frac{n-h}{m-h})$ comparisons, where one comparison is for ranking the pivots and $\frac{n-h}{m-h}$ comparisons are for splitting the non-pivot items. Since each comparison is replaced by a call of BC with confidence $1 - \frac{\delta_0}{1 + \frac{n-h}{m-h} + T_2}$, by the union bound, with probability at least $1 - \delta_0$, all these calls of BC return correct results. Finally, since the call of QS uses at most $|A_\tau|^2$ comparisons and each comparison is replaced by a call of BC with confidence $1 - \frac{\delta_0}{|A_\tau|^2}$, by the union bound, QS returns the correct result with probability at least $1 - \delta_0$. Therefore, BKS-FRF returns the top-$k$ items of $S$ with probability at least $1 - 3\delta_0 = 1 - \delta$. This proves the correctness.

**Sample complexity.** By Theorem 2, MQSelect conducts $O(\frac{n}{m})$ comparisons in expectation except the call of QS. In BKS-FRF, each comparison is replaced by a call of BC with confidence $1 - \frac{\delta_0}{1 + \frac{n-h}{m-h} + T_2}$, which conducts $O(\log \frac{1 + \frac{n-h}{m-h} + T_2}{\delta_0}) = O(\log \frac{n \log m}{m \delta_0})$ comparisons. Thus for BKS-FRF, Line 2 conducts $O(\frac{n}{m} \log \frac{n \log m}{m \delta_0})$ comparisons in expectation. For the call of QS, its expected sample complexity is $\mathbb{E}[O(|A_\tau|)]$. Each comparison of QS is replaced by a call of BC with confidence $1 - \frac{\delta_0}{|A_\tau|^2}$, and thus, by similar steps as the proof of Lemma 16 (we only need to change $m$ to $h + 1$ in the new steps), Line 3 conducts $O(|A_\tau| \log \frac{|A_\tau|}{\delta_0}) = O(\frac{n}{m} \log \frac{n}{m \delta_0})$ comparisons in expectation. Therefore, by $\delta_0 = \frac{\delta}{3}$, the expected sample complexity of BKS -FRF is $O(\frac{n}{m} \log \frac{n \log m}{m \delta})$. This proves the sample complexity, and the proof of Lemma 17 is complete. $\square$