# OpenReview forum: "Sample Complexity Bounds for Active Ranking from Multi-wise Comparisons"
_NeurIPS.cc/2021/Conference — NeurIPS 2021 Poster_

### Official Review · Reviewer_7QRR · 2021-07-12

**Rating:** 7
**Confidence:** 4

**Summary:**

This paper studies the problem of active ranking from ($m$) multi-wise comparisons. The focus on past literature was on pairwise comparisons $m=2$. For this problem the authors considered both  top-$k$ items selection and full ranking. For both problems the authors provide theoretical guarantees (lower and upper bounds) on the sample complexity, i.e., the number of comparison. Furthermore, contrary to previous related works, the current model is not parametric and quite general.


**Ethical Concerns:**

No.

**Limitations And Societal Impact:**

Yes.

**Main Review:**

The paper is well-written, interesting, and motivated. The introduction is comprehensive, and the model is clear. I read all the proofs and everything seems okay -- some proofs are quite standard and straightforward. The algorithms are also reasonable, some of which are motivated by the special case of pairwise comparisons. To me, this paper is a nice contribution, and therefore I recommend accepting the paper.

I do not have any important comments or suggestion except the following:

For the non-deterministic setting (or, noisy) it is assumed that one can repeat the same comparisons; simple averaging then allows one to "get rid" of the noise, at the price of an additional logarithmic factor due to repetition. The underlying main assumption here is, however, that the comparison responses statistically independent and identical. I suspect that this assumption is quite strong, and in practice, the responses will be quite dependent if not the same. For example, in crowdsourcing platforms there is no reason to believe that humans over these platforms will provide independent responses. Therefore, in my opinion the non-deterministic is less motivated. The remedy for this might be: 1) to assume some kind of dependency over the repeated responses, 2) limit the number of allowed repetitions, 3) consider a model where repetitions is not allowed at all. For the last two cases, some kind of structure must be induced for the problem to be feasible.

It would be great if you could discuss/mention in the conclusion open and challenging questions, future directions, etc., that follow from your work.

**Time Spent Reviewing:**

48

---

> ### Author Response · Authors · 2021-08-10
> **To Reviewer 7QRR**
>
> Thank you very much for your review and constructive comments, which helped us significantly improve the quality of this paper.  Our detailed point-by-point responses are as follows:
>
> > **Your Comment:** For the non-deterministic setting (or, noisy) it is assumed that one can repeat the same comparisons; simple averaging then allows one to "get rid" of the noise, at the price of an additional logarithmic factor due to repetition. The underlying main assumption here is, however, that the comparison responses statistically independent and identical. I suspect that this assumption is quite strong, and in practice, the responses will be quite dependent if not the same. For example, in crowdsourcing platforms there is no reason to believe that humans over these platforms will provide independent responses. Therefore, in my opinion the non-deterministic is less motivated. The remedy for this might be: 1) to assume some kind of dependency over the repeated responses, 2) limit the number of allowed repetitions, 3) consider a model where repetitions is not allowed at all. For the last two cases, some kind of structure must be induced for the problem to be feasible.
> It would be great if you could discuss/mention in the conclusion open and challenging questions, future directions, etc., that follow from your work.
>
> **Our Response:**
>
> 1) The independence of comparisons is a widely adopted assumption in the literature, but we agree that from some perspective, especially from a Bayesian perspective, the independence may not always hold. We want to clarify that the independence of comparisons means a conditional independence given the values of the hidden parameters. For instance, when a coin is fixed, then the toss results of the coin are all independent samples from a fixed Bernoulli distribution. However, if we do not use the conditional probabilities, then the results of different tosses can be dependent. For crowd-sourcing, we can view the average preferences of the users as fixed hidden parameters. For each comparison, the learner samples a random user with replacement and asks this user about its preference, and thus, the comparisons are independent given the values of the hidden parameters. In this example, a comparison is correct means its result is consistent with the majority of the users. We will clarify the above point in the final version. Thanks for pointing out this ambiguity.
>
> 2) Assuming dependence of samples or limiting the number of samples may make the ranking problems unsolvable or infeasible. For instance, for two items $i$ and $j$ where all the comparison results are determined by the first three comparisons, if the first three comparisons are noisy, we may not be able to order these two items with high confidence. Thus, assuming dependence of samples or limiting the number of samples are interesting but require careful formulation, which are open problems requiring future investigations and deserve an independent paper in its own right.

---

> > ### Comment · Reviewer_7QRR · 2021-08-25
> > **Response to response**
> >
> > Thanks much for the feedback. Indeed, assuming dependence needs a careful formulation, and is possible only if some structure is induced on the data, but I appreciate that this is out of the scope of the current submission.

---

### Official Review · Reviewer_hWKg · 2021-07-20

**Rating:** 6
**Confidence:** 4

**Summary:**

The paper systematically analyzes the sample complexity of top-k selection and ranking under active multi-wise comparisons. It studies the winner model and full ranking model of multi-wise comparison, both deterministic and with 2/3 correct probability. The algorithms are mainly based on an extension of either QuickSelection or QuickSort.

**Ethical Concerns:**

The paper is mostly theoretical.

**Limitations And Societal Impact:**

The paper is mostly theoretical.

**Main Review:**

The paper contains comprehensive results about various cases of active ranking. While the results look good and are quite complete, I feel that they can still be improved in several ways:

i) What are the implications of the results? More discussion will be helpful to understand the results. Generally, it seems like m-way comparison can help by a $O(\log m)$, $O(m)$ or $O(m \log m)$ factor. Why can it be different?

ii) What techniques are involved? Previous methods also use QuickSelect or QuickSort to solve ranking but under different assumptions. Are the algorithms presented related to previous algorithms?

In general, I think the results are good but more discussion can make the paper better.

**Time Spent Reviewing:**

3

---

> ### Author Response · Authors · 2021-08-10
> **To Reviewer hWKg**
>
> Thank you very much for your review and constructive comments, which helped us significantly improve the quality of this paper.  Our detailed point-by-point responses are as follows:
>
> > **Your Comment:** The paper contains comprehensive results about various cases of active ranking. While the results look good and are quite complete, I feel that they can still be improved in several ways:
> i) What are the implications of the results? More discussion will be helpful to understand the results. Generally, it seems like m-way comparison can help by a O(\log{m}), O(m), or O(m\log{m}) factor. Why can it be different?
> ii) What techniques are involved? Previous methods also use QuickSelect or QuickSort to solve ranking but under different assumptions. Are the algorithms presented related to previous algorithms?
> In general, I think the results are good but more discussion can make the paper better.
>
> **Our Response:**
> 1.1 The differences of the reductions in terms of $m$ come from three main reasons. First, the winner feedback provides less information than the full-ranking feedback when $m > 2$ but provides the same information when $m = 2$. Specifically, for $m > 2$, a winner feedback only returns the best item of the compared ones, while a full-ranking feedback returns the full ranking of the compared items which inherently contains the information about the best item. Thus, ranking from the full-ranking feedback should provide no worse reductions compared to ranking from the winner feedback. Second, finding the top-$k$ items mainly needs the information about the best items, while finding the full ranking needs the information about all the items. Thus, under the winner feedback setting, when $m$ increases, the information provided to the top-$k$ selection problem may increase faster than that provided to the full ranking problem, leading to different reductions. Third, when the comparisons are noisy, the problems of how to utilize the $m$-wise comparisons become more complex and we need more comparisons to reduce the influence of the noise. Thus, the reductions in the sample complexities may increase at a lower speed compared to ranking from noiseless comparisons.
>
> 1.2 Main results and implications. This work (partially) answers and gives significant insights to the questions whether and to what degree using multi-wise comparisons can reduce the sample complexities for finding the top-$k$ items or the full ranking of a set of items. Our algorithms and results can also inspire the future studies on multi-wise ranking under other settings. For top-$k$ selection and full ranking from noiseless comparisons, we have already achieved optimal sample complexity upper bounds and lower bounds (ignoring constant factors). For top-$k$ selection from noisy comparisons, there are only loglog gaps (assuming $k$ is not large) between our upper bounds and lower bounds, which are almost constant for most practical cases.
>
> 2 . To the best of our knowledge, previous works that used Quick-Sort or Quick-Select all focused on the pairwise cases. We are the first to generalize these two classical algorithms to the multi-wise cases. The generalizations are not trivial as we need to consider many careful designs (see response to Reviewer Qykc). For instance, in MQSelect, we need to split the items to $m$ piles but not split them into two piles as in pairwise algorithms Quick-Sort and Quick-Select.  This change also incurs a much more complex analysis of the algorithms’ sample complexities compared to the classical pairwise versions. Another key difference is that the multi-wise algorithms need to call the pairwise versions but not themselves recursively like the pairwise versions. Otherwise, the sample complexity would be higher. We also extend these two classical algorithms to ranking from noisy comparisons, while they were only designed for ranking from noiseless comparisons. In the extension, we also need careful designs like setting the complicated values of $\delta_1$ in BKS and BKS-FRF appropriately.

---

> > ### Comment · Reviewer_hWKg · 2021-08-25
> > **Thank you for the response**
> >
> > Thank you for the response!

---

### Official Review · Reviewer_Qykc · 2021-07-28

**Rating:** 5
**Confidence:** 4

**Summary:**

This paper considers the problem of ranking from multi-wise comparisons where the goal is either to find the set of top-k items or to find a complete ranking over the items. The paper considers a model for multi-wise comparisons where the feedback is either-- (1) deterministic where the correct result is returned; or (2) probabilistic where the feedback is correct with probability larger than 1/2. The paper assumes that there is an underlying ordering over items and considers two models of feedback-- (1) winner feedback where the best item in a set is returned; (2) full-ranking feedback where a complete ordering over arms is returned. The paper proposes lower and upper bounds on sample complexity for each of these 8 problem settings and outlines the reduction in sample complexity due to the use of multi-wise comparisons.


**Limitations And Societal Impact:**

Broader impact section is not present.

**Main Review:**

Modeling Assumption: The probabilistic model for multi-wise comparisons proposed in this paper does not seem motivated by practical applications. As the value of m increases, it seems unrealistic to assume that the best arm (or correct answer) will be returned with a probability of 1/2 + \Delta. One should expect the best arm to have win probability much less than 1/2 for larger values of m. Furthermore, the proposed model differs significantly from other well-studied models such as MNL/Random Utility Models as win probability of the best arm here is a function of the other arms in the set. It is not clear why this model is interesting to study. Since the reduction in sample complexity due to m can be dependent on the underlying model, it is not clear how these results generalise to more realistic problem settings.

Technical Novelty: Most of the results in this paper are simple extensions of existing results from theoretical computer science for both the deterministic and probabilistic settings. The idea to use multi-way pivoting in the Quick Select/Sort algorithm is a natural extension of the usual two-way pivoting. There are other classic algorithms that can be similarly extended to this setting, for example, one can construct m-way heaps and perform Heap Select/Sort over these m-way heaps by using m-wise winner feedback. The probabilistic setting is also based on existing/similar ideas except that one needs to perform the comparisons with high confidence over a sufficiently small instance (for example, see Feige et al., 1994). Finally, the lower bounds also do not require any new ideas.

Overall, I do not think that this paper satisfies the expectations from a theory paper at NeurIPS.


**Time Spent Reviewing:**

6

---

> ### Author Response · Authors · 2021-08-10
> **To Reviewer Qykc**
>
> Thank you very much for your review and constructive comments, which helped us significantly improve the quality of this paper.  Our detailed point-by-point responses are as follows:
>
> > **Your Comment:** 1. Modeling Assumption: The probabilistic model for multi-wise comparisons proposed in this paper does not seem motivated by practical applications. As the value of m increases, it seems unrealistic to assume that the best arm (or correct answer) will be returned with a probability of 1/2 + \Delta. One should expect the best arm to have win probability much less than 1/2 for larger values of m. Furthermore, the proposed model differs significantly from other well-studied models such as MNL/Random Utility Models as win probability of the best arm here is a function of the other arms in the set. It is not clear why this model is interesting to study. Since the reduction in sample complexity due to m can be dependent on the underlying model, it is not clear how these results generalise to more realistic problem settings.
>
> **Our Response:**
>
> 1) We agree that typically it is more practical that when the number of arms/items increases, the probability that a comparison returns the correct result tends to decrease. However, there are also practical cases for which the probability that the comparison returns a correct result stays in a range with increasing value of $m$. For example, when someone chooses a favorite movie, restaurant, or candidate from multiple options, the correct probabilities of these comparisons may always be larger than $\frac{2}{3}$ when $m$ increases from 2 to 10 or a not large number. In this case, we can directly apply our algorithms, upper bounds, and lower bounds, and enjoy the reductions in the sample complexities. The comparisons may also refer to querying an expert, who may also provide a correct probability larger than $\frac{1}{2} + \Delta$ or $\frac{2}{3}$ for various values of $m$.
>
> 2) Our $\frac{1}{2}+\Delta$ m-wise comparison model can also be justified in many scenarios that use an iterative subroutine to conduct m-wise comparisons over a set in a smaller time-scale. Here, we give a simple example. We use $p_{i,S}$ to denote the probability that item $i$ wins the comparison performed on set $S$ (assuming $|S| \leq m$) and we use $i^*$ to denote the best item of $S$. If $p_{i^*,S} \geq p_{i,S} + \Delta$ for any item $i \neq i^*$ and any set $S$, then by repeatedly comparing $S$ for $O(\frac{1}{\Delta^2}\log|S|)$ times, one can find the best item of $S$ with confidence $\frac{2}{3}$. We can use the above procedure as a subroutine in each iteration of our algorithms, and get the algorithms for this case while only introducing an additional $\log{m}$ factor to the sample complexities. In more general cases, we can use similar tricks but the additional factors may vary. For instance, in the setting where only the best $\log|S|$ items of a set $S$ may win the comparisons over $S$, then by using the similar procedure as above, only a $\log\log{m}$ additional factor needs to be added to the sample complexities. When the value $\log|S|$ is changed to other values, the additional factor will also change, hence, it is not easy to find a closed-form expression for the sample complexity in general cases. We will add the above insights in the revised version of this paper.
>
> 3) Moreover, in addition to the case you describe (that is, all comparisons return correct results with probability $\frac{1}{2} + \Delta$), in the paper we also investigate the case when the comparisons are always correct. These results also form an important contribution of the paper. This model is appropriate for applications for which the noise is negligible. For instance, the comparisons may refer to asking an expert to do a query on the items, where the noise may be negligible.
>
> 4) Further, note that even the parametric models that have been widely adopted may also be impractical in various ways. These parametric models may even have stronger assumptions on the comparison probabilities (as we only require the probabilities to be in a certain range while the parametric models require them to be certain values). For the problems under the parametric models, it is also unclear how the sample complexities would change when the practical applications do not perfectly fit the models (i.e., robustness of the parametric models). Thus, while we agree that most models are an imperfect representation of the real world, the ones proposed in the paper (as well as the parametric models described above) provide useful abstractions of real systems.
>
> > **Your Comment:** 2. Technical Novelty: Most of the results in this paper are simple extensions of existing results from theoretical computer science for both the deterministic and probabilistic settings. The idea to use multi-way pivoting in the Quick Select/Sort algorithm is a natural extension of the usual two-way pivoting. There are other classic algorithms that can be similarly extended to this setting, for example, one can construct m-way heaps and perform Heap Select/Sort over these m-way heaps by using m-wise winner feedback. The probabilistic setting is also based on existing/similar ideas except that one needs to perform the comparisons with high confidence over a sufficiently small instance (for example, see Feige et al., 1994). Finally, the lower bounds also do not require any new ideas.
>
> **Our Response:**
>
> 1) Using an $m$-wise heap will not get the same upper bound as in our work. To construct an $m$-wise heap, we need $O(\frac{n}{m}\log_m{n})$ comparisons and taking out the top item for $k$ times needs $O(k\log_m{n})$ comparisons. Thus, the total sample complexity will be $O((\frac{n}{m}+k)\log_m{n})$, $\log_m{n}$ higher than the upper bound $O(\frac{n}{m} + k)$ in our work. In fact, with our thorough investigations and comprehensive literature search, using generalizations of Quick-Select or Quick-Sort is the only way we have found so far that can provide such reductions or sample complexities in our work.
>
> 2) Our algorithms were carefully designed to obtain the results that we achieved and were not obvious and simple extensions. There are many important questions that necessitate new and innovative techniques. For example, in algorithm MQSelect, why and how should we split the algorithms into $m$ piles rather than some other number of piles? This question also involves a much more complex analysis of the algorithms’ sample complexities compared to the classical pairwise versions. Similarly, in algorithm MQSelect, why do we call the pairwise algorithm Quick-Select but not do a recursive call on MQSelect itself as in the classical algorithm Quick-Select? In algorithms BKS and BKS-FRF, why do we set the value of $\delta_1$ to be such complicated numbers? All of these are not found in the literature and require careful designs and mathematical analysis, without which, we could not have achieved these results.
> 3) Please note that generalizing classical algorithms has been a tried-and-true approach to obtain promising results, and should in no way diminish the novelty of the work. First, building on classical ideas to solve new problems has been very common. For example, most of the bandit algorithms generalize from the two classical algorithms UCB and Thompson Sampling, while still resulting in interesting and significant results. In fact, modifying classical algorithms and bringing them to a new area inject new insights to new areas. Further, to the best of our knowledge, we are the first to extend these classical algorithms to multi-wise settings.
> 4) Lower bounds. For top-$k$ selection and full ranking from noiseless comparisons, our lower bounds are already optimal (ignoring constant factors) and there is no need to use more complex methods or techniques. For top-$k$ selection from noisy comparisons, there are only loglog gaps between our upper bounds and lower bounds (assuming $k$ is not large), which are almost constant for most practical cases. For full ranking from noisy comparisons, finding the lower bound is much more complex. One reason is that the structures of the wrong results returned by the comparisons may also influence the sample complexity, and we have not found a worst instance to formulate better lower bounds.
>
> **Other motivations.**
>
> 1. Our work is a generalization of the problem of ranking from pairwise comparisons and attempts to answer the natural but important question: “Can we save comparisons by comparing more than two items at a time?” We have made significant progress for fully understanding this key question in the paper.
>
> 2. Our model can be viewed as the best case one can achieve by using $m$-wise comparisons, and thus, providing significant instructions and insights. The models studied in this work are the simplest cases for learning the ranking. Therefore, if one cannot get an $f(m)$ sample complexity upper bound in our model, then it is unlikely that one can get an $f(m)$ or better upper bound in other settings. Also, the sample complexity lower bounds for the noiseless cases are hard lower bounds for all settings.

---

> ### Author Response · Authors · 2021-08-27
> **Further Response to Reviewer Qykc**
>
> Dear Reviewer Qykc,
>
> Based on your comments on modeling assumptions and the technical novelty of our paper, we have tried our best to address your concerns. Please see our response last time.
>
> Since there are only a few days left in the discussion stage, could you kindly check and see whether our responses have clarified your concerns satisfactorily?
>
> Thanks so much again for your careful review and constructive comments for our paper, which have helped improve our work significantly!
>
> Best,
> Authors

---

### Decision · Program_Chairs · 2021-09-27

**Decision:**

Accept (Poster)

**Comment:**

This paper considers the problem of ranking from multi-wise comparisons where the goal is either to find the set of top-k items or to find a complete ranking over the items. The paper considers a model for multi-wise comparisons where the feedback is either-- (1) deterministic where the correct result is returned; or (2) probabilistic where the feedback is correct with probability larger than 1/2. The paper assumes that there is an underlying ordering over items and considers two models of feedback-- (1) winner feedback where the best item in a set is returned; (2) full-ranking feedback where a complete ordering over arms is returned. The paper proposes lower and upper bounds on sample complexity for each of these 8 problem settings and outlines the reduction in sample complexity due to the use of multi-wise comparisons. There is a near consensus amongst reviewer that this is a "border-line" work.